



# Aerosol particle characteristics measured in the United Arab Emirates and their response to mixing in the boundary layer

Jutta Kesti[1], John Backman[1], Ewan J. O'Connor[1], Anne Hirsikko[1], Eija Asmi[1], Minna Aurela[1], Ulla Makkonen[1], Maria Filioglou[2], Mika Komppula[2], Hannele Korhonen[1], and Heikki Lihavainen[1,3]

[1]Finnish Meteorological Institute, Helsinki, 00560, Finland
[2]Finnish Meteorological Institute, Kuopio, 70211, Finland
[3]Svalbard Integrated Arctic Earth Observing System, Longyearbyen, 9170, Norway

**Correspondence:** Jutta Kesti (jutta.kesti@fmi.fi)

**Abstract.** Aerosol particles play an important in role in the microphysics of clouds and hence on their likelihood to precipitate. In the changing climate already dry areas such as the United Arab Emirates (UAE) are predicted to become even drier. Comprehensive observations of the daily and seasonal variation in aerosol particle properties in such locations are required reducing the uncertainty in such predictions. We analyse observations from a one-year measurement campaign at a background location in the United Arab Emirates to investigate the properties of aerosol particles in this region, study the impact of boundary layer mixing on background aerosol particle properties measured at the surface and study the temporal evolution of the aerosol particle cloud formation potential in the region. We used in-situ aerosol particle measurements to characterise the aerosol particle composition, size, number and cloud condensation nuclei (CCN) properties, in-situ $SO_2$ measurements as an anthropogenic signature and a long-range scanning Doppler lidar to provide vertical profiles of the horizontal wind and turbulent properties to monitor the evolution of the boundary layer. Anthropogenic sulphate dominated the aerosol particle mass composition in this location. There was a clear diurnal cycle in the surface wind direction, which had a strong impact on aerosol particle total number concentration, $SO_2$ concentration and black carbon mass concentration. Local sources were the predominant source of black carbon, as concentrations clearly depended on the presence of turbulent mixing, with much higher values during calm nights. The measured concentrations of $SO_2$, instead, were highly dependent on the surface wind direction as well as on the depth of the boundary layer when entrainment from the advected elevated layers occurred. The wind direction at the surface or of the elevated layer suggests that the cities of Dubai, Abu Dhabi and other coastal conurbations were the remote sources of $SO_2$. We observed new aerosol particle formation events almost every day (on four days out of five on average). Calm nights had the highest CCN number concentrations and lowest $\kappa$ values and activation fractions. We did not observe any clear dependence of CCN number concentration and $\kappa$ parameter on the height of the daytime boundary layer, whereas the activation fraction did show a slight increase with increasing boundary layer height, due to the change in the shape of the aerosol particle size distribution where the relative portion of larger aerosol particles increased with increasing boundary layer height. We believe that this indicates that size is more important than chemistry for aerosol particle CCN activation at this site. The combination of instrumentation used in this campaign enabled us to identify periods when anthropogenic pollution from remote sources that had been transported in elevated layers was present, and had been mixed down to the surface in the growing boundary layer.



# 1 Introduction

Aerosol particles have an important role in many processes in the atmosphere, such as the hydrological cycle (Ramanathan et al., 2001). Unforeseen changes in the global water cycle are the greatest threats in the changing climate (Wehbe et al., 2018; Wehbe and Temimi, 2021). The areas that are already very dry are predicted to dry even more in the future and water scarcity will cause crises in the less developed countries (Stocker et al., 2014). Aerosol particles moderate cloud properties so that an abundance of aerosol particles lengthens the cloud lifetime and hence change cloudiness and rainfall patterns on Earth (Albrecht, 1989; Jiang et al., 2006). Aerosol particle-cloud processes and interactions are still poorly understood, which complicates forecasting of changes in the rainfall patterns in the future climate (Stocker et al., 2014).

Several studies suggest aerosol particles have a net cooling effect on climate (Stocker et al., 2014), but estimates of the effect of aerosol particles on rainfall vary widely (Dave et al., 2017; Fan et al., 2018). Rosenfeld et al. (2019) stated that the interaction between aerosol particles and clouds overestimates the cooling effect in present climate models and that a positive forcing, possibly through deep clouds, has been excluded from the total effect in the models.

The size of the aerosol particles and hence their direct impact on scattering of solar radiation depends on the amount of water that is bound with the aerosol particles, termed hygroscopicity (Köhler, 1936; Svenningsson et al., 1994), with more hygroscopic aerosol particles able to bind more water. Hygroscopicity depends on the aerosol particle size and chemistry (Saxena et al., 1995), and on the environment and aerosol particle history (Hitzenberger et al., 1997), with aerosol particles having high hygroscopicity acting more easily as cloud condensation nuclei (CCN) and hence likely to form cloud droplets. The aerosol particle hygroscopicity can be described in terms of the $\kappa$ parameter, which for atmospheric particulate matter typically ranges between 0.1–0.9 (Fitzgerald et al., 1982; Hudson and Da, 1996; Dusek et al., 2006). Previous studies show that aerosol particle size is more important than composition when concerning aerosol particle CCN activation (Junge and McLaren, 1971; Dusek et al., 2006). Hitzenberger et al. (1997) found that aerosol particle hygroscopicity was size-dependent, with larger aerosol particles being more hygroscopic. This size-dependence was also observed in several other studies (Ye et al., 2013; Jaatinen et al., 2014; Sarangi et al., 2019).

There are many natural and anthropogenic sources of aerosol particles, which can produce aerosol particle populations with different compositions and size distributions. Knowledge of the aerosol particle population, particularly the size distribution, is therefore necessary when attempting to constrain the climate impact of aerosol particles. Very few previous studies on aerosol particle properties and transport in the Arabian Peninsula region exist (Semeniuk et al., 2014, 2015). Lihavainen et al. (2016) used in situ observations to study aerosol particle properties in western Saudi Arabia, finding a clear seasonal variation in $PM_{10}$ and $PM_{2.5}$ concentrations with the highest concentrations occurring from February to June. The seasonal variation was related to the frequency of dust events in the area. They also observed maximums of monthly-averaged total aerosol particle number concentrations of around $13{,}000~\mathrm{cm^{-3}}$ from August to September, and that the total number concentration was dominated by frequent new particle formation events. Using a multi-wavelength PollyXT lidar, Filioglou et al. (2020) observed multiple



elevated aerosol particle layers at a site in the United Arab Emirates, with significant transport from Saudi Arabia, Iran and

Iraq. The night-time residual layers they observed contained mixtures of mineral dust and urban-marine aerosol particles.

In this study we measured the properties of aerosol particles over one full year at a background site in the United Arab Emirates. In situ aerosol particle measurements were used to obtain the aerosol particle size distribution and composition at the surface and Doppler lidar measurements were used to investigate the role of vertical mixing, both for lofting local sources away from the surface, and for the entrainment of elevated aerosol particle layers into the boundary layer and down to the

surface. A description of the site and measurement setup are presented in Section 2. In Section 3, measurements are analysed in the context of the meteorological conditions in the region. We present both the diurnal and seasonal variation of aerosol particle properties, including composition, identify new particle formation events, and determine how these properties changed in different boundary layer mixing conditions. We use case studies in representative conditions to highlight the impact of vertical transport.

## 70  2  Instrumentation and methods

### 2.1  Site description

The properties of atmospheric aerosol particles were measured at a background site at a palm tree farm ($25°14'7.8''$ N, $55°58'39.97''$ E, 165 metres above sea level, Filioglou et al., 2020). The Arabian Gulf and the city of Dubai with a population of around 3.2 million (Dubai Statistics Center) are about 70 km west from the site. The Gulf of Oman is 40 km to east

from the site. The surroundings are mainly sand desert with some agricultural settings (Wehbe et al., 2017). The surroundings of the station represent rural background without major local pollution sources.

The measurement campaign was conducted from February 2018 to February 2019. The in-situ instruments were housed in a modified 20 foot sea container. The container was next to a main farm building that was used as a weekend home and there were only minor activities at the farm during the campaign. The temperature inside the container was kept stable, at around

25 °C, with two air conditioning units. A Doppler lidar (HALO Photonics) was situated on the top of the container in order to have as open a horizon for scanning as possible. A vertically-pointing PollyXT Lidar was placed next to the container. Both lidars were protected from the direct sunlight with sun shades.

The meteorological observations were averaged hourly, with a 1-hour data coverage of 94 %. The highest hourly temperature of 48 °C was measured in July 2018 and the lowest hourly temperature of 10 °C was measured in February 2019. The hourly

averaged maximum wind speed measured at the station was $17.5 \text{ m s}^{-1}$ in May 2018. The minimum relative humidity of 6 % was measured in March 2018 and the maximum, 90 %, in November 2018. The measured ambient pressure was stable during the measurement period (minimum 972.6 hPa, maximum 1004.2 hPa). Rain rarely reached the surface, usually evaporating as it fell into dry layers below the cloud level - a frequent characteristic for rainfall events in this region (Wehbe et al., 2019, 2020); only eight rain events were observed at ground level during the campaign. The events were very short, lasting less than

one hour each.





Daily and annual variation of meteorological conditions during the campaign year are shown in Figure 1. The highest temperatures were measured at noon during the summer months when the relative humidity also reached its minimum. The highest wind speeds were measured in early summer. The daily variation of the wind direction at the site was very distinctive, following a similar pattern every day, turning from the eastern direction (Gulf of Oman) to the northwest (Arabian Gulf) during

the day, and then back to the east after the sun had set. From March to August the variation between these two directions were more pronounced than from September to February. The wind speeds at the surface were in general quite low, averaging only $2.2\,\mathrm{m\,s^{-1}}$. The wind speed showed a clear daily pattern, with the nights typically very calm, and stronger winds during daytime peaking around noon.

## 2.2 In-situ measurements

The air was sampled through a main inlet at height of 4.5 m above the ground, with a flow rate of $16.7\,\mathrm{l\,min^{-1}}$ and a cut-off nozzle of $PM_{10}$. The air sample was dried with a Dekati diluter (dilution factor of 8.02). In this system, the dilution air is compressed, dried (dew point $-40\,°C$) and filtered air. After the dilution unit, which was located outside the container, the sample air was taken through a silica gel drying unit which acted as a backup in case of compressor failure. This system kept the relative humidity well below 40 % as recommended (Baltensperger et al., 2003, WMO/GAW Report 153). The aerosol particle

losses in this kind of dilution system have been reported before by Giechaskiel et al. (2009) who stated that the ejector dilutor does not change the characteristics of the aerosol particle size distribution and that the aerosol particle losses are negligible.

The aerosol particle size distribution was measured with a Differential Mobility Particle Sizer (DMPS). The DMPS consists of a 28-cm-long Hauke-type differential mobility analyser (DMA) (Winklmayr et al., 1991) with a closed-loop sheath flow arrangement (Jokinen and Mäkelä, 1997) and a Condensation Particle Counter (CPC), TSI model 3772. The measured dry

aerosol particle size range is from 7 to 800 nm, which is divided into 30 discrete bins. The DMPS data inversion was performed as described by Aalto et al. (2001) and Wiedensohler et al. (2012).

A Cloud Condensation Nuclei counter (CCNc) was operated in five different supersaturations (0.1, 0.2, 0.3, 0.6, 1.0). Each supersaturation measurement cycle was set to take 10 minutes so one complete cycle took 1 hour. The full cycle includes an additional supersaturation of 0.09 to allow the temperatures to drop and stabilise after measuring at a supersaturation of 1.0. The

CCNc system also incorporated one extra feature that is not part of the standard CCNc system provided by the manufacturer; the additional feature enabled the CCNc to measure the fraction of activated aerosol particles as a function of size by size selecting aerosol particles with a DMA (10–250 nm size range). To determine the fraction of activated aerosol particles at a certain size, the CCNc number concentration was compared to the number concentration of a CPC that was measuring in parallel with the CCNc. These scans were performed one supersaturation at a time from 10 nm to 250 nm (size ramp took

10 min) and then moving on to the next supersaturation to begin the next scan until all supersaturations were completed. The CCNc alternated between total CCN concentration and size selected CCN concentration so that every second supersaturation sequence was with a size selecting DMA in front so that each supersaturation cycle took 1 hour to complete.

The $SO_2$ concentrations were measured with a Thermo Scientific 43i-TLE using a 30 s time resolution. The sample air was taken through a separate Teflon tube inlet. The aerosol particle absorption coefficient was measured with an aethalometer





(AE-31, Magee Scientific) at seven wavelengths (370, 470, 520, 590, 660, 880 and 950 nm) at 1 hour time resolution. The list
of instruments used in the measurement campaign is given in Table 1.

### 2.3    Offline sampling with filters

The $PM_{2.5}$ samples were collected with an automatic outdoor station (Tecora Skypost PM HV) operating at a flow rate of
$16.67\,l\,min^{-1}$. The sampler automatically loaded a new filter every 96 h. There were 15 filters loaded in one cassette. The first
filter was used as a blank filter and loaded and unloaded to the sampling position without any air flow. Aerosol particles were
collected on pre-combusted 47 mm quartz fiber filters (Tissuquartz, PALL Life Science, Ann Arbor, USA). A total of 52 filters
provided samples, with 6 filters providing blanks; the averaged concentrations from the blank filters were subtracted from the
sample concentrations. The average blank concentrations of organic carbon (OC) and elemental carbon (EC) were 0.58 and
$0.04\,\mu g\,m^{-3}$, which were on average 19 and 4 % of the sample concentrations, respectively. The average blank concentrations
of analysed ions were below $0.02\,\mu g\,m^{-3}$, which were on average below 5 % of the sample concentrations.

Punches of $1\,cm^2$ were taken from the filters and analysed for OC and EC using thermal-optical techniques, which are based
on the evolution of carbon species at different temperatures (Birch and Cary, 1996; Watson et al., 2005; Bauer et al., 2009). In
the first phase, a sample is heated in steps from 550—900 °C using the EUSAAR-2 protocol (Cavalli et al., 2010) in an inert
Helium atmosphere to remove OC. The organics may be pyrolysed to pyrolysed carbon (PC) during the inert phase, and this is
observed by a decrease in the laser signal monitoring the transmittance through the sample matrix. In the second phase, oxygen
is introduced and the temperature is elevated step-wise as described by Cavalli et al. (2010). Carbon is oxidised to $CO_2$, which
is then converted to methane and detected by the Flame Ionisation Detector. The PC formed during the temperature programme
is compensated by determining the split point when the laser signal returns to its original value before pyrolysis. Methane was
used as an internal standard and sucrose as an external standard.

Another filter punch ($1\,cm^2$) was used for analysing the main water-soluble inorganic ions (chloride, nitrate, sulphate,
sodium, ammonium, potassium, magnesium and calcium) using an ion chromatographic method based on the standard SFS-
EN 16913:2017 (Ambient air. Standard method for measurement of $NO_3^-$, $SO_4^{2-}$, $Cl^-$, $NH_4^+$, $Na^+$, $K^+$, $Mg^{2+}$, $Ca^{2+}$ in $PM_{2.5}$
as deposited on filters). The filter sample was put in a test tube, extracted with 10 ml of ultrapure water for 30 minutes in an
ultrasonic bath, and then directly filtered to the sampling vial using a syringe filter (Acrodisc 13, porosity 0.45 $\mu m$). Anions
and cations were analysed using two identical ion chromatographs (Waters Corporation), which consisted of a separation
module (Alliance 2695: an autosampler, an injector and a pump) and a conductivity detector (Waters 432). In the anion ion
chromatograph, calibration standards and samples were injected through the guard colum (Waters Anion Guard-Pak) and
analytical column (Waters IC-Pak A HR 4.6 x 75 mm, Waters) using borate-gluconate eluent and in the cation system through
a guard column (Metrosep C4 Guard 5 x 4.0, Metrohm) and analytical column (Waters IC-Pak C M/D; 3.9 x 150 mm) using
an $EDTA/HNO_3$ eluent. The uncertainties of the analytical methods were 10 % or better for all analysed anions and cations.



## 2.4 Data processing

The data were first quality checked and screened for outliers, such as extreme high or low concentrations, arising from an instrument malfunction or instrument maintenance. The aerosol particle data reported here are in standard temperature and pressure conditions (273.15 K, 1013.25 hPa).

The CCN data was processed in two parts. One part was the size-selected aerosol particles which were detected by both the CCN and a CPC concurrently to determine at which sizes the aerosol particles could activate into cloud drops. The activation diameter where 50 % of the aerosol particles had been activated was determined by fitting the curve

$$y = -\left(\frac{a + (b - a)}{(1 + 10^{((c - D_p) * d))}}\right) \tag{1}$$

to the ratio of CCN to CPC aerosol particle number concentration as a function of size. The curve fitting was done after
manually checking the size distribution spectra and, for a successful curve fit, the following criteria had to be met: (1) The activation curve should only once cross the 0.5 (50 %) line, (2) The curve should plateau out at higher diameters and roughly remain between 0.9 and 1.1.

The second part of the CCN data was the total number concentration of CCN at five different supersaturations. These data were aggregated into 10-minute averages as that was the duration of the respective supersaturation measurement periods. This
means that, for a particular supersaturation, the total CCN number concentration is represented by 10 minutes of data every second hour.

Aerosol particle water content and the conditions for cloud droplet activation can be predicted using the $\kappa$-Köhler theory (Petters and Kreidenweis, 2007). The value of $\kappa$ describes an aerosol particle's hygroscopicity and hence the aerosol particle chemistry, with more hygroscopic aerosol particles having a higher $\kappa$ value. When supersaturation increases, $\kappa$ decreases
because smaller aerosol particles are also activated to become CCN. The value of $\kappa$ for the CCNc data was calculated at four different supersaturations: 0.2, 0.3, 0.6 and 1.0, using

$$\kappa = \frac{4A^3}{27D_c^3 \ln^2 S_c}, A = \frac{4\sigma_w M_w}{RT\rho_w}, \tag{2}$$

where $D_c$ is the critical diameter, $S_c$ is the supersaturation, $\sigma_w$ is the surface tension of water, $M_w$ is the molecular weight of water, $R$ is the universal gas constant, $T$ is the temperature and $\rho_w$ is the density of water. A typical range of $\kappa$ for atmospheric
aerosol particles extends from about 0.1 to 0.7, corresponding to nearly hydrophobic to very hygroscopic aerosol particle types (Petters and Kreidenweis, 2007).

The Mass Absorption Cross-section (MAC) values used for black carbon concentration calculation were [39.53, 31.11, 28.13, 24.79, 22.16, 16.62, 15.39] m² g⁻¹ (Arnott et al., 2005). The multiple scattering enhancement correction factor (C0) value 3.5 was used based on World Meteorological Organization recommendations (GAW Report No. 227; http://wmo-gaw-
wcc-aerosol-physics.org/ wmo-gaw-reports.html). The aethalometer measurement is known to suffer from filter loading and

scattering artifacts. There are a number of different methods for correcting these artifacts, and here, the approach presented by Virkkula et al. (2007) was chosen.

The $SO_2$ concentration data was first averaged hourly. After averaging, the data was corrected based on measured zero levels during the campaign (Komppula et al., 2017).

All the results are presented in local time UTC + 4 hours.

## 2.5   Halo Doppler lidar measurements

The Halo Photonics Streamline Doppler lidar is a pulsed lidar operating at 1565 nm using heterodyne detection (Pearson et al., 2009). The instrument has full hemispheric scanning capability and provides range-resolved profiles of the backscattered signal intensity, in terms of signal-to-noise ratio (SNR), and radial Doppler velocity. For this campaign, the instrument was operated

with a radial resolution of 30 m covering a range from 90 m to 9600 m. Additional instrument technical specifications and configuration are given in Table A1. The scan schedule selected for the campaign comprised vertical stare mode with 3 s integration time interspersed with a set of velocity-azimuth-display (VAD) scans every 15 minutes at three different elevations from horizontal: 15, 45 and 70 degrees. Each VAD scan contained 24 rays equally spaced in azimuth with an integration time of 2 s per ray, except for the highest elevation scan at 70 degrees, which had an integration time of 3 s per ray.

Background correction of SNR was performed as described in Manninen et al. (2016) and Vakkari et al. (2019) in order to obtain reliable uncertainty estimates for SNR and radial Doppler velocity (Rye and Hardesty, 1993). Profiles of calibrated attenuated backscatter coefficient were then derived from the vertical profiles of corrected SNR using the telescope function determined with the method of Pentikäinen et al. (2020) and the dissipation rate of turbulent kinetic energy calculated from the variability in the vertical Doppler velocities (O'Connor et al., 2010).

The vertical profile of horizontal wind was calculated from the VAD scans (Päschke et al., 2015) with the lower elevation scans providing better vertical resolution closer to the surface. The horizontal wind calculation assumes homogeneity, which may not be valid under strongly turbulent conditions, and the lower elevation scans also suffer less from possible violation of this homogeneity assumption.

    The presence of turbulent mixing is diagnosed from the dissipation rate (O'Connor et al., 2010), and the combination of

attenuated backscatter coefficient, vertical velocity skewness, dissipation rate, horizontal wind, and vector wind shear are used to derive a boundary layer classification (Manninen et al., 2018). This boundary layer classification identifies the mixing layer height and also identifies which regions of mixing are connected to the surface. During the daytime, mixing connected to the surface is assumed to be convective (buoyancy production), but there are also cases where there is mixing at night connected to the surface arising from wind shear. For our purposes we assigned *daytime mixing* associated with convection to periods of

mixing observed between 5 am and 8 pm and *nighttime mixing* to periods of mixing observed between 8 pm and 5 am.


## 3    Results and discussion

### 3.1    Aerosol particle properties

Chemical and physical properties of aerosol particles were investigated by statistical means. The statistics for daily means of different measured aerosol particle properties are shown in Table 2. The mean aerosol particle total number concentration was
$4812 \, \mathrm{cm}^{-3}$, which was much lower than the mean total concentration of $10630 \, \mathrm{cm}^{-3}$ measured by Lihavainen et al. (2016) in Saudi Arabia. We also compare our measurements to those made in another arid region in South Africa. Vakkari et al. (2013) observed median concentrations of 1856 and $7805 \, \mathrm{cm}^{-3}$ in the size range of 12–840 nm at Botsalano and Marikana in South Africa, and Laakso et al. (2012) observed a mean aerosol particle total number concentration of $6310 \, \mathrm{cm}^{-3}$ in the size range of 10–840 nm in the polluted Highveld area in South Africa, where several coal-fired power plants were located around the
measurement site. The total number concentration for nucleation mode aerosol particles (10–25 nm) was $753 \, \mathrm{cm}^{-3}$, for Aitken mode aerosol particles (25–100 nm) $2436 \, \mathrm{cm}^{-3}$ and for accumulation mode aerosol particles (100–800 nm) $1623 \, \mathrm{cm}^{-3}$. The mean $SO_2$ concentration was 0.53 ppb ($1.4 \, \mu \mathrm{g} \, \mathrm{m}^{-3}$), which was consistent with the mean $SO_2$ concentrations of $2.6 - 4.2 \, \mu \mathrm{g} \, \mathrm{m}^{-3}$ measured by Al Katheeri et al. (2012) in Al Mirfa, UAE, during 2007–2009. The mean black carbon concentration measured during this campaign was $1.48 \, \mu \mathrm{g} \, \mathrm{m}^{-3}$, and was lower than the mean concentration of $2.1 \, \mu \mathrm{g} \, \mathrm{m}^{-3}$ observed in Saudi
Arabia by Lihavainen et al. (2016). The observed CCN number concentration was highest when measuring at a supersaturation of 1.0 ($3029 \, \mathrm{cm}^{-3}$), with the largest $\kappa$ parameter of 0.52 at a supersaturation of 0.3.

### 3.1.1    Aerosol particle chemical components

Chemical analysis indicated that sulphate had the highest contribution (around 50 % of the analysed mass) to aerosol particle composition in the UAE (Table 3, Fig. 2) and it was mainly in a form of ammonium sulphate (($NH_4$)$_2SO_4$). The contribution
of sea salt to $SO_4^{2-}$ was small (up to 4 %) at the site, indicating that the majority of the sulphate was from anthropogenic origin. Sulphate and $SO_2$ concentrations were correlated with each other, having higher concentrations in spring and summer. The annual average sulphate mass concentration in the UAE was high ($10.34 \, \mu \mathrm{g} \, \mathrm{m}^{-3}$). There are two oil refineries located on the eastern and western side of the measurement site: Fujairah oil refinery around 40 km in the east and Sharjah oil refinery around 70 km in the west. The refineries are assumed to be the main reason for the measured high sulphate mass concentration
at the site. When compared to other studies, the annual average sulphate mass concentration in the UAE was over twice as high as concentrations measured in rural South Africa sites and five times higher than concentrations measured in an urban area in USA (Aurela et al., 2016; Chan et al., 2018). On the other hand, Wu and Wang (2007) measured one and a half times higher 24-h filter sulphate mass concentration in a suburban area 45 km northwest of the centre of Shanghai and over two times higher concentration measured in a mountainous area 50 km north of the centre of Beijing. The carbonaceous components (OC + EC)
made the second highest contribution to aerosol particle mass. EC forms during incomplete burning and its origin is always primary whereas OC can be primary or secondary. A rough guide to the origin of the carbonaceous aerosol particles is given by the ratio between OC and EC concentrations, if only one major EC source (traffic or biomass combustion) is dominant (Turpin and Huntzicker, 1995). The OC to EC ratio was 2.81, which resembles values measured at urban background areas (e.g. Aurela





et al., 2011). EC and BC showed a strong correlation, and, on average, the BC to EC ratio was 2.2. We also observed calcium,
magnesium, sodium and chloride, which are typically observed in the coarse aerosol particle size fraction (> 2.5 $\mu$m). These
ions mostly originate from soil and sea salt. The ratio of cations and anions was close to one, so the ion balance was neutral.
Potassium showed higher concentrations during May 2018, which may be a result of biomass burning. However, a similar
increase was not seen for BC.

### 3.1.2   Daily and annual variation

Figure 3a shows that the total aerosol particle number concentration usually peaked after local noon. This diurnal maximum
is dominated by frequently observed secondary new particle formation (NPF) events that had a clear diurnal cycle, together
with a potential change in aerosol particle population due to the change in wind direction during the day. The total aerosol
particle number concentration was lowest during late summer and early autumn, and relatively constant during the rest of the
year. Figure 4 shows aerosol particle total number concentration separated into nucleation, Aitken and accumulation modes.
Nucleation mode aerosol particles have the highest concentration at noon whereas accumulation mode aerosol particles have
their highest concentrations during night. This is an expected result because accumulation mode aerosol particles are aged
aerosol particles that have the longest lifetime in the atmosphere and are trapped in the nighttime stable layer whereas nucleation
mode aerosol particles are newly formed and grow fast to larger sizes during the day. This diurnal cycle is typical for locations
experiencing NPF events, and was also observed by Lihavainen et al. (2016) in western Saudi Arabia. The growth of the
boundary layer also impacts the diurnal cycle of the concentration in accumulation mode aerosol particles through the dilution
of the aerosol particles into a larger volume as the boundary layer depth increases.

Similar to aerosol particle number, very strong diurnal and annual cycles were observed in $SO_2$ concentrations (Fig. 3b).
The highest $SO_2$ concentrations were observed around midday, especially during spring (March to June). The wind direction
at midday was coming from west or northwest (Fig. 1d), where sources of $SO_2$ exist, such as Dubai. Lower concentrations at
other times of the year are most likely because of the change in the prevailing midday wind direction to directions where no
strong or nearby sources exist. The black carbon concentration was highest during the night (Fig. 3c) with low concentrations
during the day presumably due to the increase in mixing volume in the boundary layer and increased wind speed. Black carbon
concentration did not show any clear seasonal dependence.

The daily and annual variation in $\kappa$ parameter at supersaturations of 0.3 and 1.0 is shown in Figure 5a. The $\kappa$ parameter
decreased with SS; at $SS_{0.3}$, $\kappa$ varied from 0.17–1.16, and at $SS_{1.0}$, from 0.01–0.87, with the highest values at a supersaturation
0.3. The fraction where 50 % of aerosol particles activate into cloud droplets is called the activation diameter ($D_{50}$) and is used
when calculating $\kappa$. At $SS_{0.3}$, $D_{50}$ was $68 \pm 6$ nm, and was $39 \pm 7$ nm at $SS_{1.0}$. This result suggests that aerosol particles
at the critical diameter at $SS_{0.3}$ had different chemistry and were more hygroscopic than other aerosol particles in the region
(Sarangi et al., 2019), which means that smaller aerosol particles were less hygroscopic. The $\kappa$ parameter had higher values
during daytime, which suggests that there are more sulphate aerosol particles present from the NPF events, or that the wind
direction means there is significant transport of hygroscopic aerosol particles to the site, or sulphuric acid originating from
$SO_2$ condensates on pre-existing aerosol particles, or all affecting concurrently. During nighttime there are more BC aerosol





particles present (Fig. 3d), which are less hygroscopic. $\kappa$ showed a small decrease in early autumn, and there was a small decrease also in early spring.

The activation diameter of aerosol particles decreases when supersaturation increases, which can be seen in Fig. 5b, showing higher CCN concentrations during nighttime when there were more long-lived and larger accumulation mode aerosol particles and black carbon present (see Fig. 4 and 3c). The CCN number concentrations were highest in early summer. The CCN number concentration decreased significantly in early autumn. This may be due to a change in the environmental circumstances, i.e. a change in the prevailing midday wind direction transporting aerosol particles from different sources.

The activation fraction (CCN/CN) describes the amount of aerosol particles that are activated to CCN. Figure 5c shows the diurnal and monthly changes in activation fraction; the highest activation fractions were measured in the morning and evening when the CCN number concentration was also highest. $\kappa$ values are highest during daytime, and when we consider the diurnal variation of different aerosol particle sizes shown in Figure 4, it seems that size is more important for CCN activation than chemistry at this location. The activation fraction is smaller during the daytime because there are more small non-CCN active

aerosol particles present. The reason for high daytime $\kappa$ values may be due to the presence of hygroscopic aerosol particles such as sea salt or sulphate from anthropogenic sources that have been transported to the site by westerly winds. There was a peak in the activation fraction in early spring, after which the activation fraction smoothly decreased towards the end of the year. Activation fraction was the lowest in the beginning of the year.

The aerosol particle properties followed similar daily cycle. During late morning the wind direction typically changed from

the eastern direction to northwest. Before and during the change we observed lower aerosol particle total number concentrations compared to midday, and also lower $SO_2$ concentrations. On the other hand, the BC concentration was higher before the change in wind direction after which the wind speed started to increase. The $\kappa$ parameter was lower, but CCN number concentration and activation fraction were higher. During the afternoon, when the wind was coming from northwest, we observed a peak in aerosol particle total number concentration and in $SO_2$ concentration, whereas the BC concentration decreased, presumably

due to dilution in the deeper boundary layer. The $\kappa$ parameter was higher during the day but the CCN number concentration and activation fraction decreased. After sunset and during the night, the wind direction changed back to the eastern direction with the aerosol particle total number concentration and $SO_2$ concentration decreasing and the relative portion of accumulation mode aerosol particles increasing, BC concentration starting to increase, $\kappa$ parameter decreasing and CCN number concentration and activation fraction starting to increase.

## 3.2   Impact of the boundary layer on aerosol particle properties

### 3.2.1   Turbulent and non-turbulent boundary layers

Aerosol particle properties at our UAE measurement site showed a clear diurnal cycle. This corresponds to the strong diurnal cycle in boundary layer evolution and the wind direction observed in the Doppler lidar profiles. To investigate the impact of the boundary layer evolution and wind direction on aerosol particle properties, we separated our analysis into three classes

of boundary layer conditions using the boundary layer classification derived from Doppler lidar: daytime mixing (convective





boundary layer), night time where mixing was present, and night time with no mixing present (calm). We assume that vertical mixing will dilute the concentrations from ground level sources. When there is no mixing during the night we expect an increase in concentrations as the emissions from local sources are trapped in the surface layer with only advection providing transport at ground level and limited dilution in the vertical dimension. One source of turbulent mixing in the boundary layer at night is

low-level wind jets with significant shear (Marke et al., 2018).

The median value of the $SO_2$ concentration from each boundary layer class varied from 0.13–0.2 ppb, with the highest values during daytime mixing (Fig. A1a). This is attributed to the transport of pollution from Dubai and other urban areas (see prevailing wind direction during the day, Fig. 1d). At night, concentrations were higher when there was no mixing, because of the limited dilution in the vertical. The median black carbon concentration at the surface varied between 0.9–1.9 $\mu$g m$^{-3}$ in

different boundary layer circumstances (Fig. A1b) and was more directly connected to the presence of boundary layer mixing, being lower during the day and higher during nights with no mixing.

Figure 6a shows $\kappa$ values calculated for the three different boundary layer conditions. Median $\kappa$ values ranged from 0.4–0.6 with the highest median value and largest variability during daytime mixing. This indicates that larger and more hygroscopic aerosol particles are present during the day, whether emitted locally or transported to the measurement site. One potential source

is the elevated daytime $SO_2$ concentrations from which sulphate aerosol particles can form. A smaller $\kappa$ during nighttime is typical for anthropogenic aerosol particles (Hung et al., 2014; Cheung et al., 2020).

Figure 6b shows that the median CCN number concentration was close to $1500$ cm$^{-3}$ when there was turbulent mixing present, whether during the day or at night, and was about $2000$ cm$^{-3}$ during calm nights. The lowest concentration values, observed during the day, are when there are more nucleation mode aerosol particles present, and there are more large

accumulation mode aerosol particles present at night (Fig. 4).

Figure 6c shows activation fraction for the three different boundary layer conditions. Taken annually, an average of 40–50 % of aerosol particles were able to activate as CCN at SS = 0.3, with the highest median activation fraction being observed during turbulent conditions at night. This is probably due to the higher relative contribution of larger accumulation mode aerosol particles which are more easily activated to CCN. The reason why the activation fraction is higher during turbulent rather than

calm nights is probably due to the entrainment of the residual layer above back into the nocturnal boundary layer. The aerosol particle properties of the residual layer, which is formed from the decaying deep daytime boundary layer above the nocturnal boundary layer after sunset, evolve slowly from the aerosol particle properties of the daytime convective boundary layer. The residual layer is rich with larger aerosol particles which, if mixed down into the nocturnal boundary layer, would increase the number of CCN-activated aerosol particles observed at the surface during nighttime. Another possibility is that a calm

nocturnal layer leads to a build-up of small (non-CCN active) aerosol particles which would be more effectively scavenged in a turbulent layer. Low activation fractions during daytime are probably due to NPF events occurring during the day, generating large numbers of small and hence less easily activated aerosol particles.





### 3.2.2 Boundary layer height

We also studied the impact of daytime boundary layer height on the aerosol particle measurements made at the surface. We
took all daytime measurements for each day and binned these into four height ranges using the maximum boundary layer height
for that day. Each bin was about 250 m wide to give approximate bin top heights at 1.5, 1.75, 2 and 2.25 km (1245–1515 m,
1515–1755 m, 1755–1995 m and 1995–2265 m).

The median $SO_2$ concentration showed a dependence on maximum boundary layer height (Fig. 7a) with median values
increasing with maximum boundary layer height from 0.2–0.25 ppb up to boundary layer heights of 1.75–2 km; the highest
median value, and the greatest range in values were for boundary layer heights of 1.75–2 km. However, black carbon con-
centration (Fig. 7b) showed a slight decrease with increasing maximum boundary layer height. Median concentrations varied
between 1.0–1.1 $\mu$g m$^{-3}$ with the lowest median concentration measured in the same height range as the maximum median
$SO_2$ concentration. These results suggest that BC concentrations are dominated by local sources, whereas $SO_2$ sources are not.
The residual layer is one source of $SO_2$, which requires the boundary layer to grow and entrain this source into the boundary
layer. The other source is transport within the boundary layer, evidenced by the strong $SO_2$ concentration dependence on wind
direction.

Median CCN concentrations and $\kappa$ values (not shown) displayed little dependence on maximum boundary layer height but
the median CCN-activation fraction, which ranged from 40–45 %, did increase with maximum boundary layer height (Fig.
7c). Figure 8 shows that the aerosol particle size distribution did change with maximum boundary layer height, with number
concentrations reducing as the boundary layer height increased. The growing boundary layer dilutes the aerosol particles from
local sources into a larger volume, but at the same time, there is entrainment of aged aerosol particles in the residual layer
back into the boundary layer. The change in the shape of the size distribution explains the change in CCN-activation fraction,
so although there is no change in the CCN concentration itself, the reduction in the number of small aerosol particle sizes,
which are less likely to activate, with increasing maximum boundary layer height will increase the CCN-activation fraction
with increasing maximum boundary layer height.

### 3.2.3 New particle formation events

We defined NPF event days based on the number concentration in the 16.8 nm size bin, with a threshold value of 3000 cm$^{-3}$
(dN/dlogDp) being exceeded in order to be classified as an event day. The relative proportion of event days was very high with
178 event days from a total of 224 measurement days. The starting time of NPF events was defined using the concentrations in
375 the 10.4 nm size bin. We identify that an event has started when the concentration in this size bin increases by at least 300 par-
ticles cm$^{-3}$ (dN/dlogDp). The actual event starting time is then estimated based on the growth rate estimated visually from a
DMPS size-concentration figure where there was a very clear NPF event. Our estimate for the growth rate was approximately
6.7 nm h$^{-1}$, so we subtracted 2 hours from the identification time obtained from analysing concentrations in the 10.4 nm size
bin to derive the time when these aerosol particles were about 1 nm in size. The start hours for NPF events determined in this
380 manner are presented in Figure 9a. NPF events usually started before noon, around 9–11 am. This was compared with the time





that the boundary layer started to grow (Fig. 9b). We defined the boundary layer growth start time based on the value of the dissipation rate of turbulent kinetic energy derived from Doppler lidar, $\epsilon$, exceeding $10^{-4}$ $\mathrm{m^2\,s^{-3}}$ at a height of 225 meters. Boundary layer growth at this height usually started around 9–10 am, which is quite consistent with the NPF event starting time.

### 3.3 Case studies

The evolution of the boundary layer was very similar day–to–day at this location in UAE. Here we show three representative case studies: deep boundary layer, shallow boundary layer, and a residual layer with significantly different properties; to examine how the evolution of the boundary layer impacted aerosol particle properties. We chose to focus on conditions between midnight and 11 am to concentrate on the time period which was most similar day–to–day, and to avoid including daily dust events and other large-scale dynamical mechanisms such as sea breezes.

#### 3.3.1 Deep boundary layer

We selected 19 May 2018 as a representative deep boundary layer case study. The boundary layer height reached a maximum of 2775 meters on this day. Figure 10a shows that the surface was calm at night and the convective boundary layer mixing started at the surface around 7 am. $SO_2$ concentrations (Fig. 10b) remained low at night and then rose rapidly as the convective boundary layer grew. There was a strong peak in aerosol particle number concentration before the mixing started (Fig. 10c) and then the sign of a NPF event was seen at 9 am. The start time for the NPF event was therefore assumed to be two hours earlier, at 7 am, which was when the convective boundary layer mixing started, and the $SO_2$ concentration began to increase. During the night, wind speeds were light at the surface ($< 5$ $\mathrm{m\,s^{-1}}$) and from the south, but there was a stronger wind jet above at around 1 kilometre height (Fig. 10d) which was coming from the north-north-west (Fig. 10e), i.e. Dubai and other conurbations. Moderate turbulence arising from the wind shear associated with this wind jet can also be observed. The wind direction at the surface changed towards the north-west at 7 am, which also coincides with when $SO_2$ concentrations increased, although the surface winds remained light.

Occasional strong peaks in DMPS concentrations can be seen 5 and 8 am. At this time the boundary layer height had not yet grown significantly, so these are most likely arising local sources, and this increase is also seen in the Doppler lidar attenuated backscatter coefficient (Fig. 10f) close to the surface. Such peaks are not seen later in the day once the boundary layer has grown deeper, and hence we suspect that any local pollution source is rapidly dispersed in a deeper boundary layer. Figure 10f also shows that there are considerable aerosol particles present in the residual layer which are then entrained into the boundary layer as it grows; hence the aerosol particle number concentration at the surface remains high.

#### 3.3.2 Shallow boundary layer

The shallow boundary layer case study (21 February 2018) had a maximum boundary layer height of 1215 meters. As in the deep boundary layer case study the surface was mostly calm at night, with Figure B1a in the Supplement showing the





convective boundary layer mixing starting around 9 am. This coincides with an increase in $SO_2$ concentrations and a change in the wind direction to northwest. A peak in number concentration of smaller aerosol particles at 10 am indicates an NPF event starting at around 8 am. However, $SO_2$ concentrations were also elevated during the night, when surface winds were also from the northwest, and then reduced by 4 am as the wind direction changed to west and southwest. The wind direction had changed at the surface by 2 am but some turbulent mixing was also present from 2–3 am which may also have helped continue to mix some elevated $SO_2$ to the surface until the wind direction aloft had also changed.

Figure B1f shows again that there are considerable aerosol particles present in the residual layer which are then entrained into the boundary layer as it grows; hence the aerosol particle number concentration at the surface remains high.

### 3.3.3 Entraining a residual layer with different properties

The case study for 8 May 2018 presents a case where the residual layer has different aerosol particle properties to the surface layer. Note that at this location, nearly all days exhibit a significant residual layer. As in previous cases, the surface was calm and convective boundary layer mixing commenced around 8 am (see Supplement Fig. C1a). Surface $SO_2$ concentrations were low throughout most of the night, and the wind direction at the surface remained east or south. Surface $SO_2$ concentrations rapidly increased once convective mixing started, even though the wind direction at the surface was from the south. However, there was an elevated layer aloft advecting from northwest to north, which the boundary layer rapidly grew into after 8 am, and we attribute the increase in $SO_2$ concentration to the entrainment of this layer. This can also be seen in Fig. C1f with the elevated layer no longer visible after 10 am. Small aerosol particles were present at the surface during the night but disappeared once the convective mixing started, and $SO_2$ concentrations also began to decrease again after this layer had been entrained. Wind speeds aloft were low, so once the source had been depleted, there was no advection bringing more $SO_2$.

### 3.3.4 Aerosol particle size distribution

For the three case study days selected, Fig. 11 showed the change in aerosol particle size distribution before and after convective mixing started at the surface. There is little change in concentration in the larger sizes (diameters > 50 nm) for both the deep and shallow boundary layer cases selected but there is significant increase in the concentrations of smaller aerosol particles (diameters < 30 nm). We attribute the change in small aerosol particle number concentration to NPF which starts at a similar time to the daytime convective mixing. Note the small mode peak is at smaller sizes for the shallow boundary layer case, for which there has been less time for aerosol particle growth as the NPF event started later. Little change is seen at larger sizes because the residual layer contains aerosol particles with a similar composition to that at the surface, as it had been transported there during daytime convection the day before.

The third case study showed a remarkable difference in the aerosol particle size distributions before and after daytime convective mixing commenced. This indicates that the aerosol particles in the residual layer above had a different composition, and was responsible for the changes in number concentrations at large sizes seen at the surface once this residual layer had been entrained into the growing daytime boundary layer and mixed down to the surface. In contrast to the other case study





days, there was also a reduction in the number concentrations of small aerosol particle sizes after daytime convective mixing
had started, and observations suggest that NPF had not begun during this period.

## 4   Conclusions

We conducted a one-year campaign deploying aerosol particle in-situ and ground-based remote sensing measurements in the
United Arab Emirates to analyse the background aerosol particle properties and their changes in different boundary layer
conditions in the region. The location selected, a palm tree farm in Al Dhaid, exhibited a clear diurnal cycle in wind direction,
with surface winds from the east at night, and west or northwest during the day. The wind direction had a strong impact on
aerosol particle total number concentration, $SO_2$ concentration and black carbon mass concentration. We observed NPF events
almost every day (178 days out of 224 measurement days). The NPF events had a similar daily pattern, which correlated with
the change in wind direction, but also with the start of daytime convective mixing. The $\kappa$ parameter usually increased during
daytime, whereas the aerosol particle activation fraction usually decreased, and CCN concentrations slightly reduced. If we
consider the diurnal variation in the aerosol particle size distribution, this indicates that size is more important for aerosol
particle CCN activation than chemistry at this site.

Anthropogenic sulphate dominates the aerosol particle mass composition in this region. Black carbon concentrations clearly
depended on the presence of turbulent mixing, with much higher values during calm nights, but the depth of the daytime
convective boundary layer had a negligible impact on black carbon concentrations. This implies that black carbon originates
from sources local to the site. $SO_2$ concentrations were highly dependent on the surface wind direction or the depth of the
boundary layer, indicating that the source of $SO_2$ was remote. $SO_2$ concentrations at the surface were high when the surface
winds were from the west or northwest, or when elevated layers transported in the same direction were entrained into the
growing boundary layer. We attribute the remote sources of $SO_2$ to be the cities of Dubai, Abu Dhabi and other coastal
conurbations.
$SO_2$ concentrations were clearly connected with the NPF events observed at the site. Most NPF events started about 2 hours
after sunrise and coincided with the start of boundary layer growth and elevated $SO_2$.

Calm nights had the highest CCN number concentrations and lowest $\kappa$ values and activation fractions. We did not observe
any clear dependence of CCN number concentration and $\kappa$ parameter on the height of the daytime boundary layer, whereas
the activation fraction did show a slight increase with increasing boundary layer height. The change in activation fraction with
boundary layer height is a result of the change in the shape of the aerosol particle size distribution with boundary layer height,
with deeper boundary layers having relatively fewer numbers of aerosol particles in the smaller size range (20–50 nm). We
attribute this to the relative amount of dilution of the aerosol particles formed in NPF events.

Usually, the residual layers formed from the previous day have the same aerosol particle characteristics as that of the new
growing boundary layer, having been transported aloft in the boundary layer of the previous day. However, occasions when
there are residual layers with different aerosol particle characteristics can be seen in the surface measurements once this layer
is entrained into the growing boundary layer. The combination of instrumentation used in this campaign enabled us to identify





periods when anthropogenic pollution from remote sources that had been transported in elevated layers was present, and had been mixed down to the surface and initiated new particle formation.

*Data availability.* The data used in this work are available upon request.

*Author contributions.* JK, EOC, AH and EA planned the structure. JK, JB and HL were responsible for the measurement data. JK, JB, EOC, MA and UM processed the data. JK, HL, EOC, MA and UM performed the data analysis. JK, EOC, HL, MA and UM wrote the paper. All the co-authors were involved in the interpretation of the results and paper editing.

*Competing interests.* The authors declare that they have no conflict of interest.

*Disclaimer.* Any opinions, findings and conclusions or recommendations expressed in this material are those of the author(s) and do not
necessarily reflect the views of the National Center of Meteorology, Abu Dhabi, UAE, funder of the research.

*Acknowledgements.* This work was supported by the National Center of Meteorology, Abu Dhabi, UAE, under the UAE Research Program for Rain Enhancement Science and the Academy of Finland Flagship funding (grant no. 337552). The work of J. Kesti is funded by the Maj and Tor Nessling Foundation (Grant 202000254). We would also like to thank Timo Anttila, Siddharth Tampi and Farah Abdi for providing on-site technical support.



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

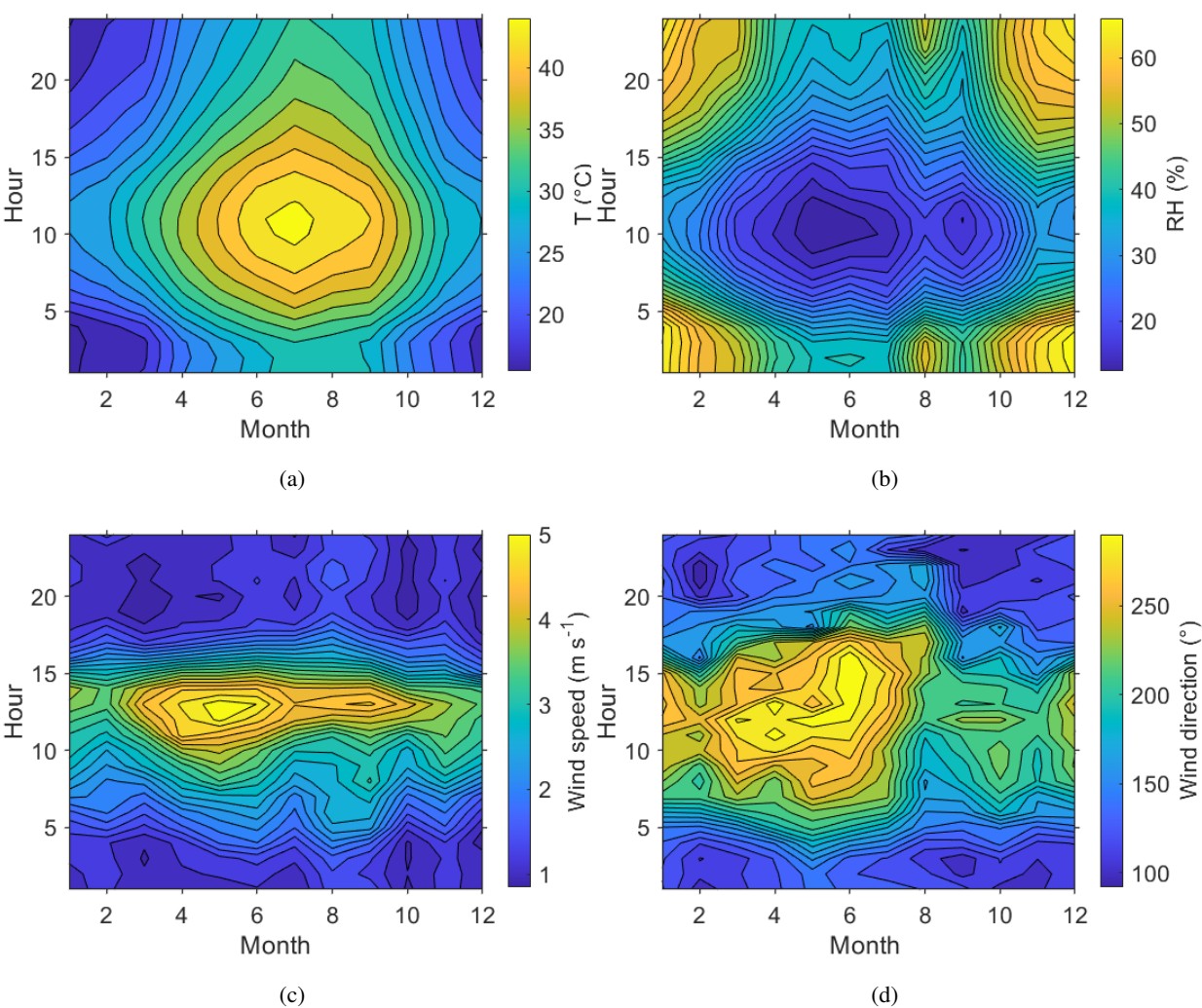

**Figure 1.** Seasonal and diurnal variation in surface a) temperature, T (°C), b) relative humidity, RH (%), c) wind speed, WS (m s$^{-1}$) and d) wind direction, WD (°). All measurements made at 7 m above ground level.



**Table 1.** Instruments operated during the measurement campaign.

| Instrument and manufacturer | Measurement parameter | Measurement details | Time resolution |
| --- | --- | --- | --- |
| Automatic Weather Station, Vaisala WXT 520 | temperature, relative humidity, precipitation, pressure, wind speed, wind direction | 7 m above ground level | 5 min |
| Differential Mobility Particle Sizer, FMI | aerosol particle size distribution | size range: 7 – 800 nm, 30 bins | 6 min 25 s |
| Aerodynamic Particle Sizer, TSI 3321 | aerosol particle size distribution, aerodynamic | size range: $0.5 - 10\,\mu$m, 52 bins | 5 min |
| Aethalometer AE-31, Magee Scientific | black carbon concentration, absorption coefficient | 7-wavelengths: 370, 470, 520, 590, 660, 880, and 950 nm | 5 min |
| Cloud Condensation Nuclei Counter, Droplet Measurement Technology, FMI | size-segregated cloud condensation concentration at different water supersaturations | size segregation: 10 – 250 nm, 20 bins, supersaturations: 0.1, 0.2, 0.3, 0.6 and 1.0 | 60 min |
| Thermo Scientific l43ITLE | sulphur dioxide concentration | The sample was taken through a separate Teflon tube inlet with an excess flow of about $10\,l\,min^{-1}$ to shorten the residence time in the tube. | 30 s |
| PollyXT LIDAR | Vertical profiles: Attenuated and aerosol particle backscatter and extinction coefficients, volume and aerosol particle linear depolarization ratio, water vapor mixing ratio, cloud and aerosol particle optical and geometrical properties | emission wavelengths: 355, 532, 1064 nm. receiver wavelengths: 355, 387, 407, 532, 607, 1064 nm. range resolution: 7.5 m | 30s |
| HALO Doppler lidar | Vertical profiles: Attenuated backscatter, Doppler velocity, dissipation rate, horizontal winds, wind shear, boundary layer classification | wavelength: 1565 nm, range resolution 30 m | 3 s, 1 min for dissipation rate, 3 min for classification, 15 min for winds and wind shear |
| Tecora Skypost PM HV | Particulate chemical composition in $PM_{2.5}$ | EC/OC | 4-day sampling time, 52 samples in the campaign period |





**Table 2.** Mean, standard deviation, 10-, 50- and 90 percentile, minimum and maximum values of daily means and data coverage for different measured variables. The size ranges for different aerosol particle modes are 10–25 nm for nucleation mode aerosol particles, 25–100 nm for Aitken mode aerosol particles and 100–800 nm for accumulation mode aerosol particles.

| Variable | Mean $\pm$ std | P $10^{th}$ | P $50^{th}$ | P $90^{th}$ | min(24 h) | max(24 h) | Data coverage (%) |
|---|---|---|---|---|---|---|---|
| N $\left[\text{cm}^{-3}\right]$ | $4812 \pm 4470$ | 1730 | 4082 | 8059 | 1594 | 14110 | 57 |
| $N_{nuc}$ $\left[\text{cm}^{-3}\right]$ | $753 \pm 1474$ | 60 | 294 | 1747 | 20 | 3251 | 57 |
| $N_{Ait}$ $\left[\text{cm}^{-3}\right]$ | $2436 \pm 3208$ | 729 | 1962 | 4226 | 652 | 11625 | 57 |
| $N_{acc}$ $\left[\text{cm}^{-3}\right]$ | $1623 \pm 1185$ | 586 | 1367 | 2957 | 300 | 3856 | 57 |
| BC $\left[\mu\,\text{g m}^{-3}\right]$ | $1.48 \pm 1.10$ | 0.54 | 1.16 | 2.85 | 0.38 | 3.43 | 72 |
| $SO_2$ [ppb] | $0.53 \pm 1.01$ | 0.10 | 0.17 | 1.26 | 0.07 | 4.00 | 72 |
| $CCN_{SS\ 0.3}$ $\left[\text{cm}^{-3}\right]$ | $2004 \pm 1849$ | 748 | 1737 | 3476 | 10 | 26211 | 62 |
| $CCN_{SS\ 0.6}$ $\left[\text{cm}^{-3}\right]$ | $2609 \pm 1674$ | 1101 | 2233 | 4279 | 61 | 16793 | 62 |
| $CCN_{SS\ 1.0}$ $\left[\text{cm}^{-3}\right]$ | $3029 \pm 1751$ | 1284 | 2629 | 5107 | 62 | 14026 | 62 |
| $\kappa_{SS\ 0.3}$ | $0.52 \pm 0.15$ | 0.33 | 0.50 | 0.72 | 0.18 | 0.92 | 21 |
| $\kappa_{SS\ 0.6}$ | $0.37 \pm 0.15$ | 0.19 | 0.36 | 0.54 | 0.10 | 0.73 | 21 |
| $\kappa_{SS\ 1.0}$ | $0.28 \pm 0.14$ | 0.12 | 0.26 | 0.47 | 0.08 | 0.58 | 19 |





**Table 3.** $PM_{2.5}$ concentrations and statistics for EC, OC and their ratio and water-soluble ions, measured in 52 4-day samples. Units for components are $\mu g\ m^{-3}$.

| Variable | Mean $\pm$ std | P $10^{th}$ | P $50^{th}$ | P $90^{th}$ |
|---|---|---|---|---|
| OC | $2.45 \pm 1.29$ | 1.18 | 2.36 | 3.77 |
| EC | $0.95 \pm 0.47$ | 0.35 | 0.91 | 1.58 |
| OC:EC | $2.81 \pm 1.03$ | 1.67 | 2.60 | 4.31 |
| $SO_4$ | $10.34 \pm 6.26$ | 4.94 | 9.48 | 20.39 |
| $NO_3$ | $0.56 \pm 0.36$ | 0.22 | 0.50 | 0.99 |
| Cl | $0.10 \pm 0.15$ | 0.03 | 0.04 | 0.24 |
| Na | $0.29 \pm 0.23$ | 0.10 | 0.25 | 0.60 |
| $NH_4$ | $2.66 \pm 1.85$ | 1.08 | 2.20 | 5.46 |
| K | $0.23 \pm 0.35$ | 0.08 | 0.14 | 0.30 |
| Mg | $0.13 \pm 0.08$ | 0.05 | 0.13 | 0.27 |
| Ca | $1.44 \pm 1.07$ | 0.48 | 1.29 | 2.25 |





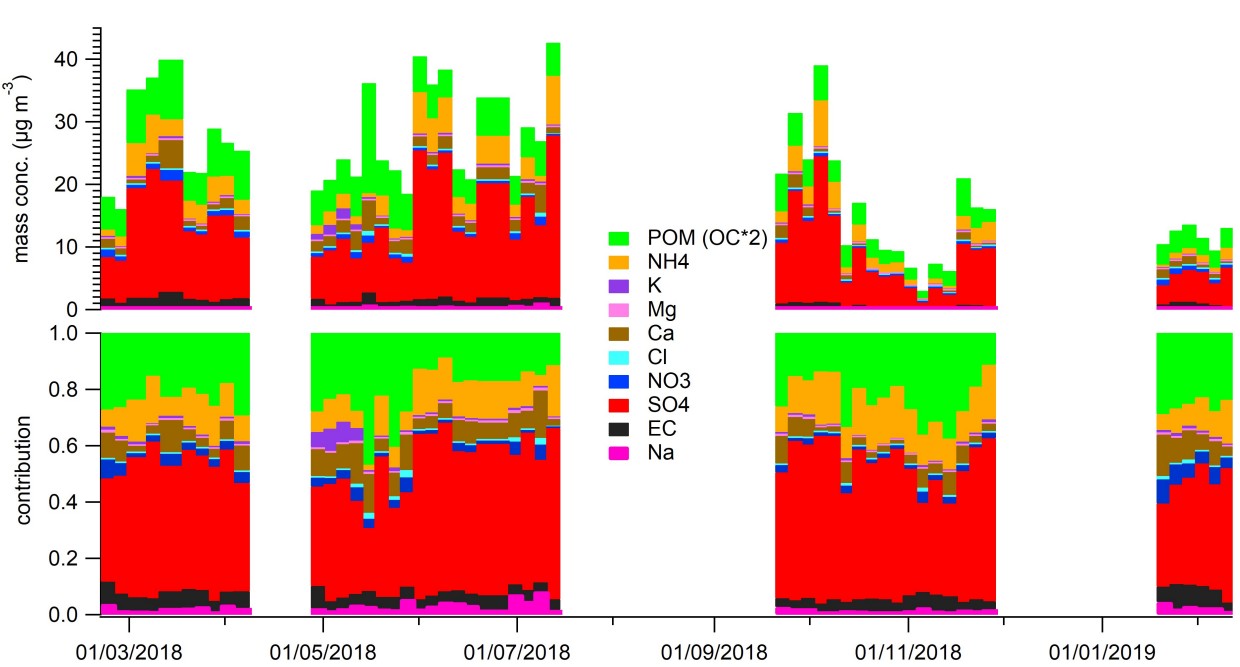

**Figure 2.** Aerosol particle chemical composition during the campaign.

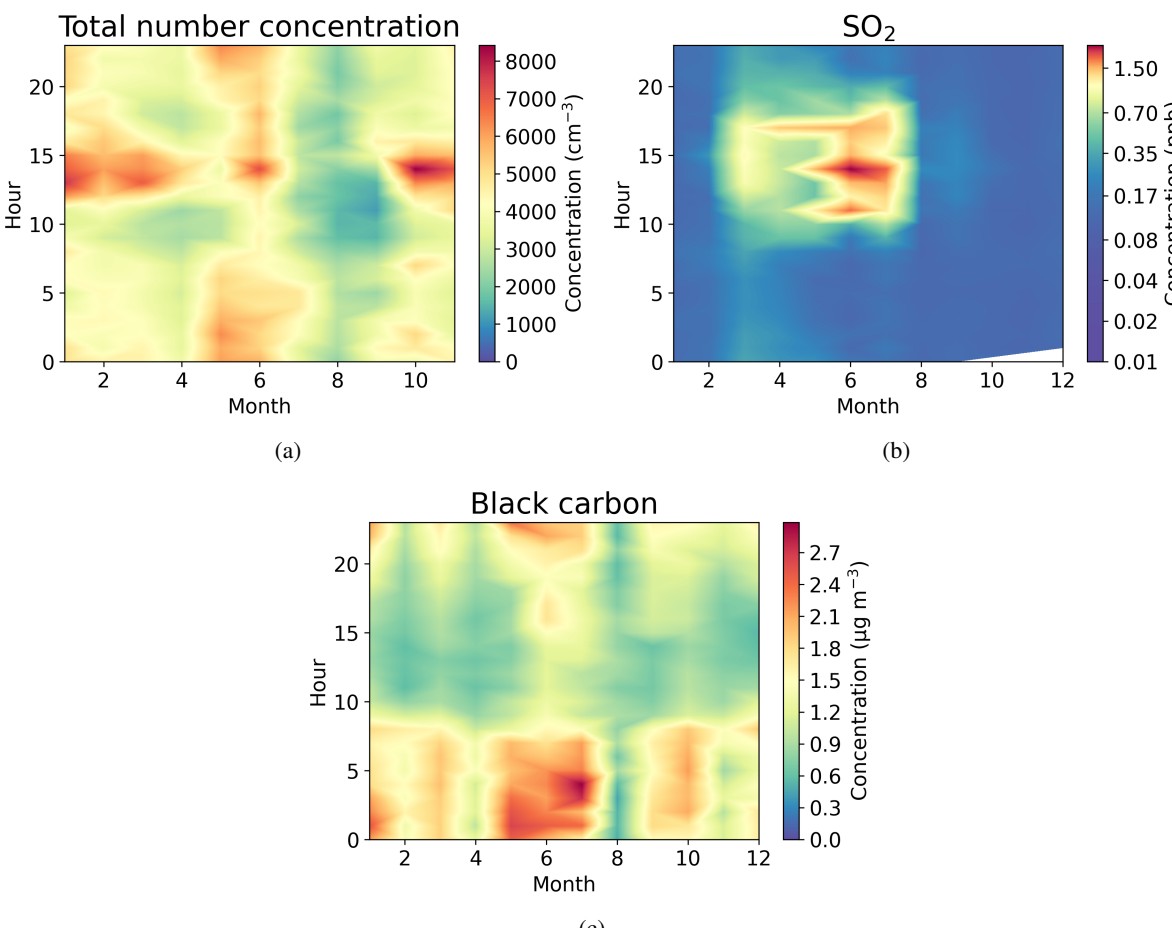

**Figure 3.** Seasonal-diurnal contour plots of a) aerosol particle total number concentration ($cm^{-3}$) measured with the DMPS, b) $SO_2$ concentration (ppb) measured with the $SO_2$ gas analyser and c) black carbon mass concentration ($\mu g\,m^{-3}$) measured with the Aethalometer. Hourly median values have been used.

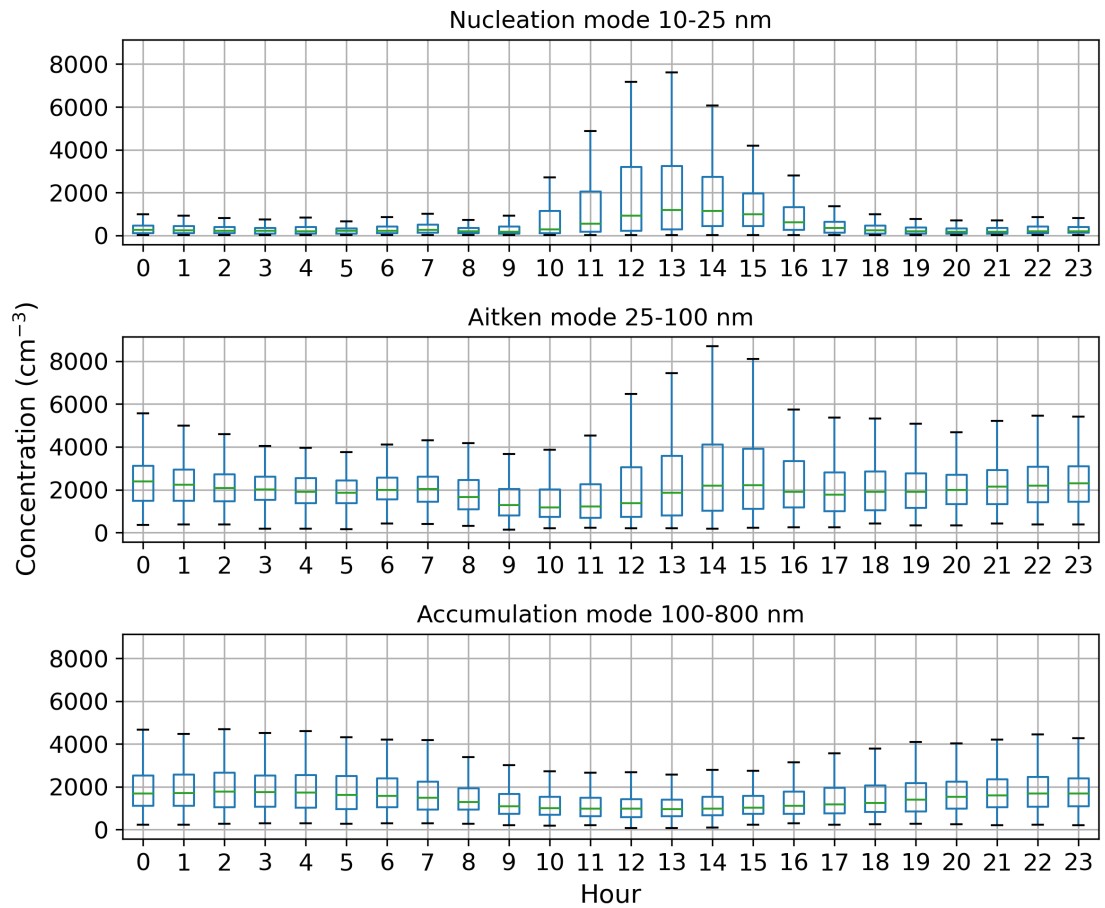

**Figure 4.** Box-and-whisker plots for hourly median concentration values ($\mathrm{cm}^{-3}$) of different aerosol particle modes. The central line (green) in each box indicates the median, and the bottom and top lines of the boxes indicate the $25^{\mathrm{th}}$ and $75^{\mathrm{th}}$ percentiles. The whiskers mark the most extreme data points that are not considered as outliers (within a range of 1.5 x interquartile range from the edges of the box).

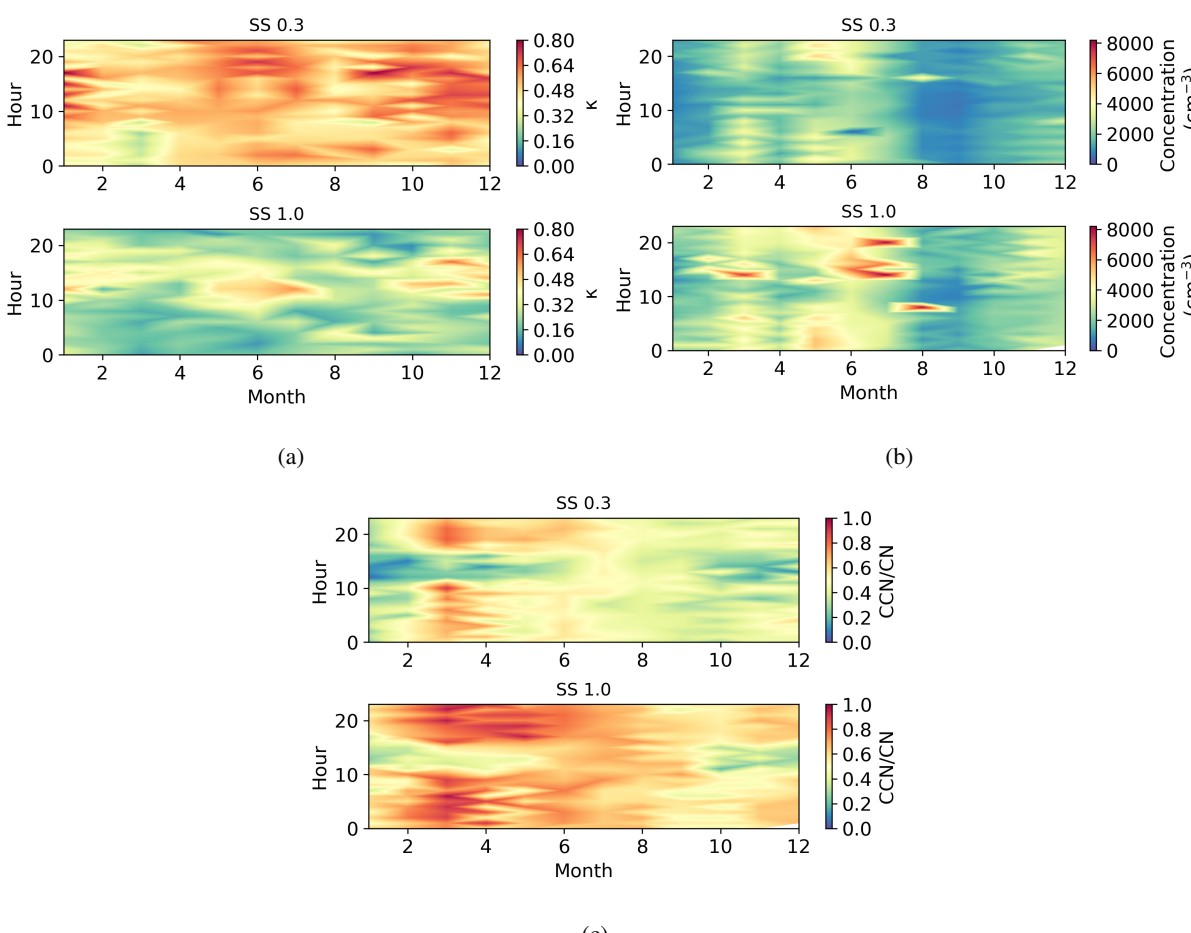

(a)

(b)

(c)

**Figure 5.** Seasonal-diurnal contour plots of a) $\kappa$ parameter, b) CCN number concentration ($\mathrm{cm^{-3}}$) and c) activation fraction (CCN/CN), in different supersaturations. Hourly median values have been used.

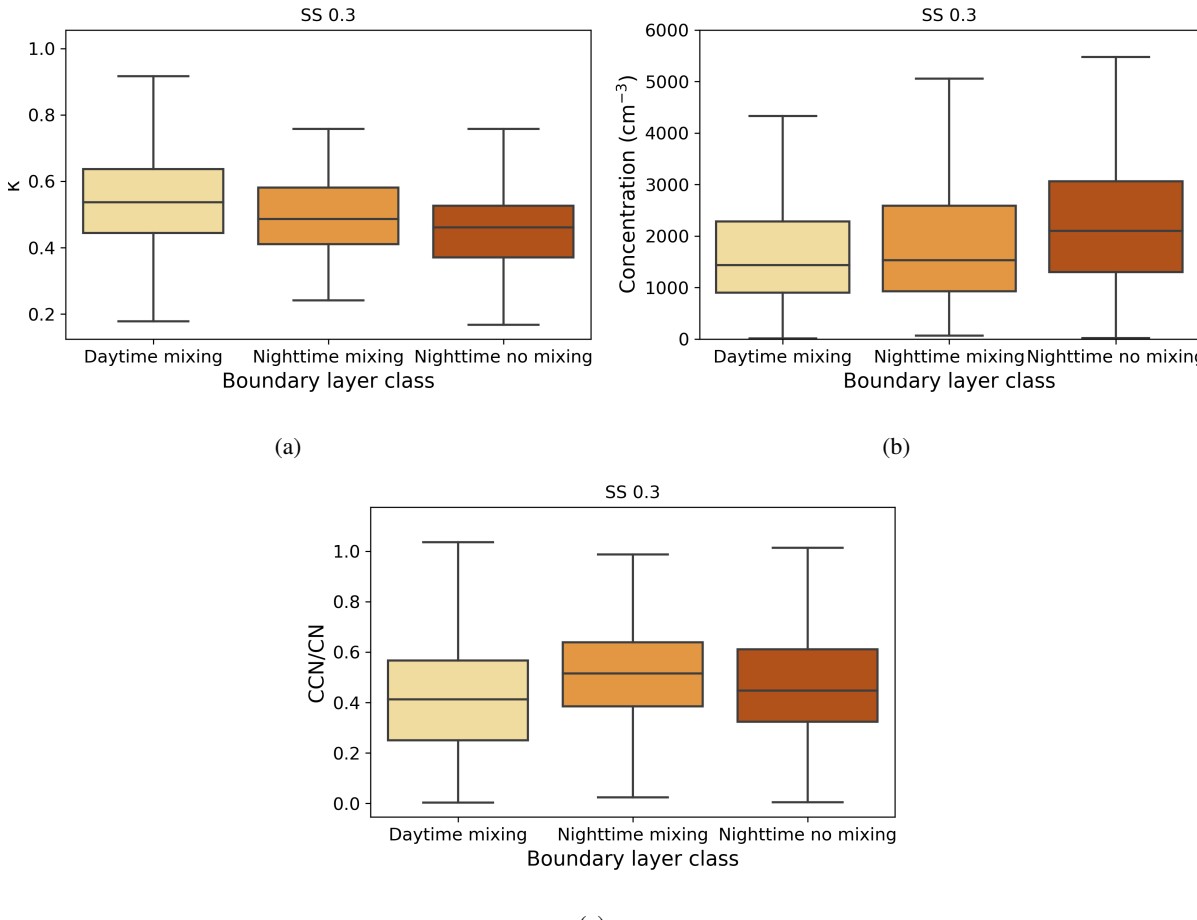

**Figure 6.** Box-and-whisker plots of hourly median values for 3 boundary layer conditions for a) $\kappa$ parameter, b) CCN number concentration ($\mathrm{cm^{-3}}$) and c) activation fraction (CCN/CN) at a supersaturation of 0.3. The central line in each box indicates the median, and the bottom and top lines of the boxes indicate the $25^{th}$ and $75^{th}$ percentiles. The whiskers mark the most extreme data points that are not considered as outliers.



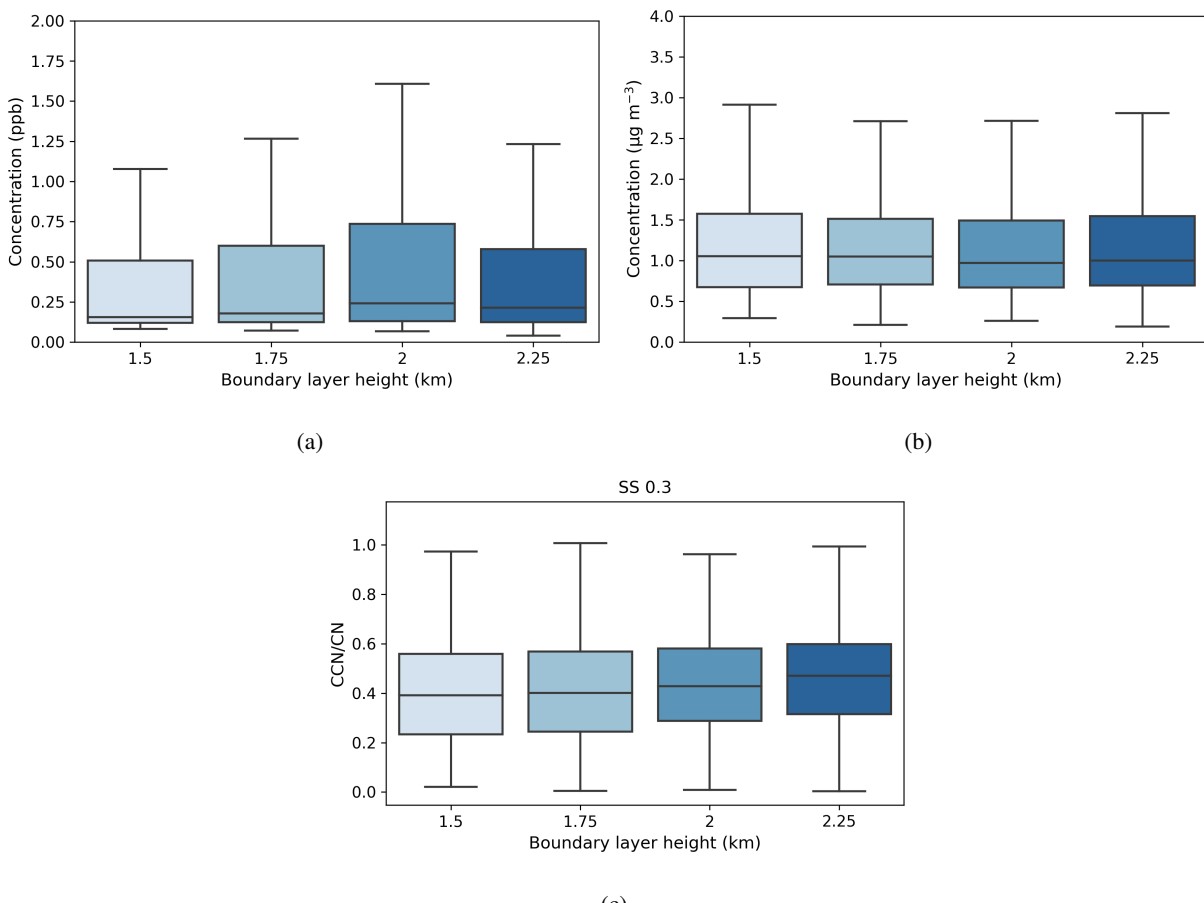

**Figure 7.** Box-and-whisker plots of hourly median values for a range of boundary layer heights for a) $SO_2$ (ppb), b) black carbon ($\mu g\ m^{-3}$), and c) activation fraction (CCN/CN) at a supersaturation of 0.3. The central line in the boxes indicate the median, and the bottom and top lines of the boxes indicate the $25^{th}$ and $75^{th}$ percentiles. The whiskers mark the most extreme data points that are not considered outliers. The boundary layer height value refers to the top of each height bin.



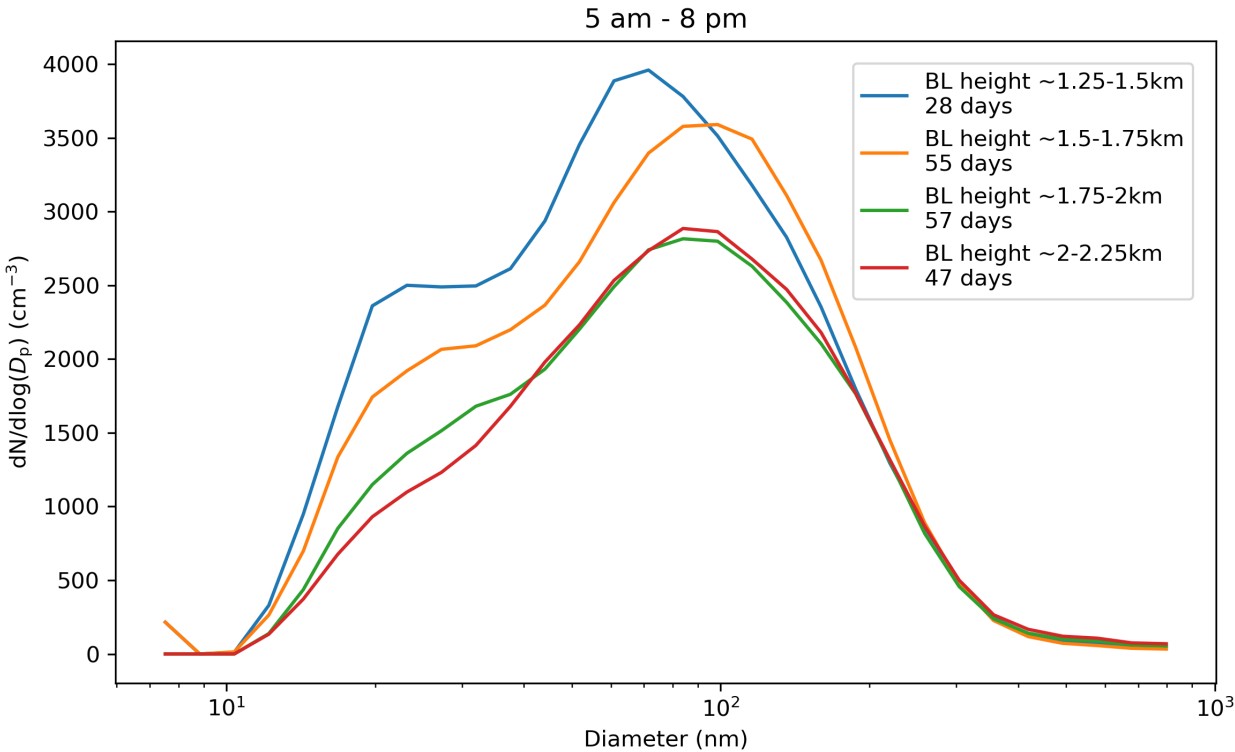

**Figure 8.** Impact of boundary layer height on the aerosol particle size distribution in $\mathrm{dN}/\mathrm{dlog}(\mathrm{d_p})$ $(\mathrm{cm}^{-3})$ measured at the surface during daytime mixing (from 5 am to 8 pm).Hourly median values are used. Number of days included in each boundary layer height range are given in the figure legend.





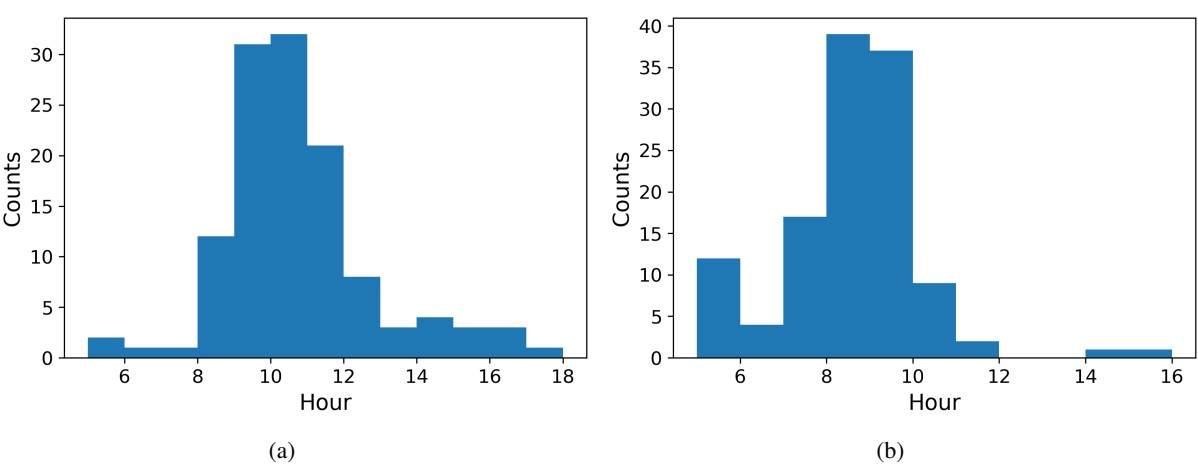

(a)                              (b)

**Figure 9.** Histograms of start time for a) NPF events and b) boundary layer growth. The time axis is local time (hours UTC+4).

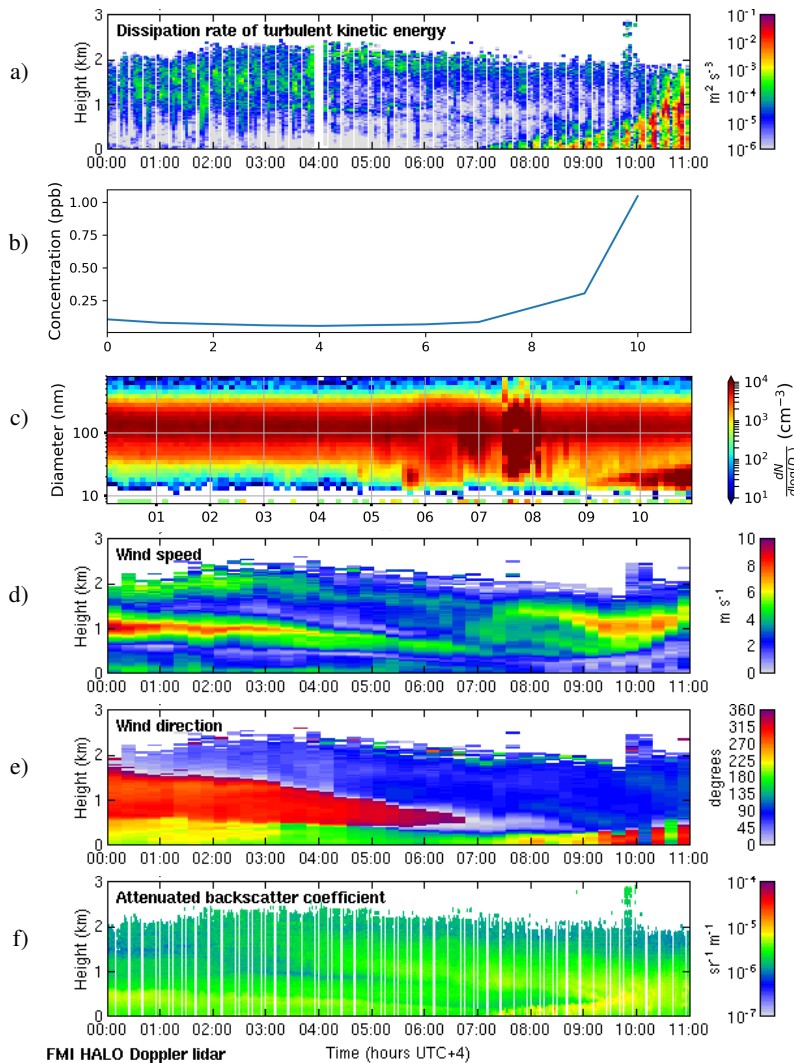

**Figure 10.** Deep boundary layer case on 19 May 2018. Panel a) displays a time-height plot of dissipation rate of turbulent kinetic energy, $\epsilon$, b) $SO_2$ concentration (ppb) measured at the surface, c) aerosol particle size distribution measured at the surface (the color indicates the concentration $dN/d\log(d_p)$ in $cm^{-3}$), and time-height plots of d) wind speed, e) wind direction, and f) attenuated backscatter coefficient. For all panels, the time axis is local time (hours UTC+4). The measurements in panels a), d), e) and f) were obtained from the HALO Doppler lidar.

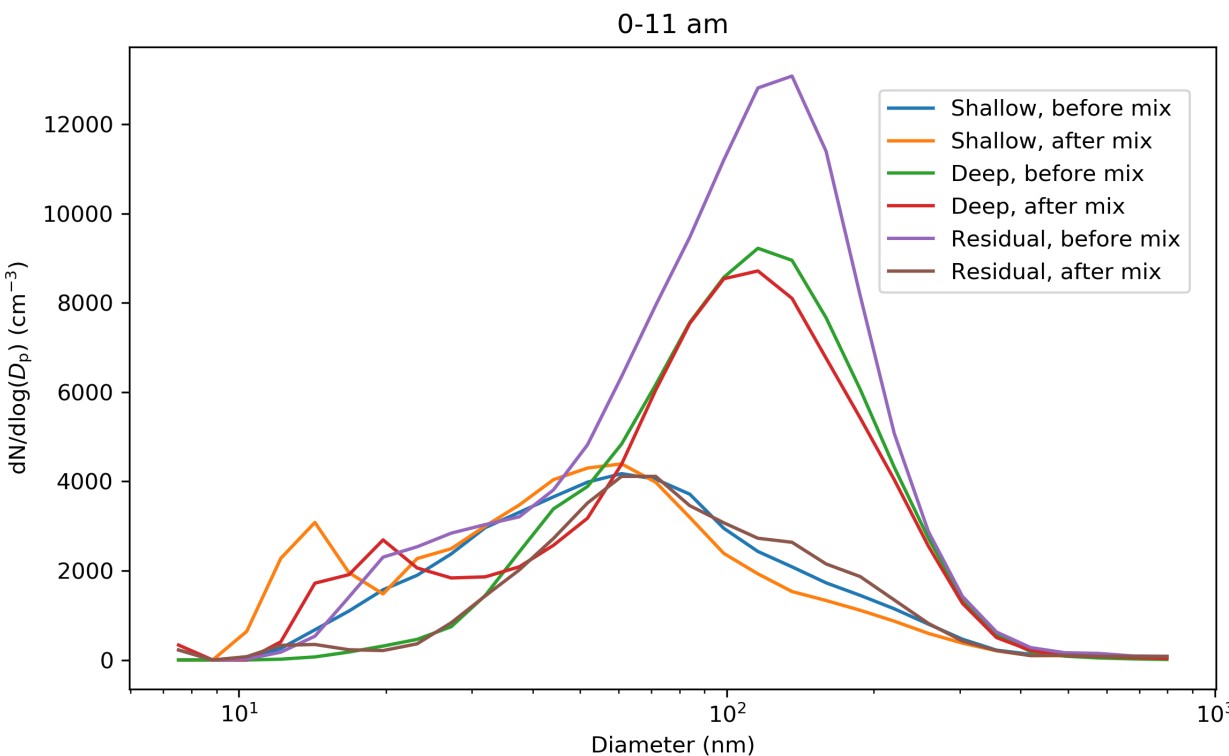

**Figure 11.** Impact of convective mixing on the aerosol particle size distributions measured at the surface for three case study days. Hourly median values are used. Aerosol particle size distributions are averaged from midnight to start of convective mixing, and from start of convective mixing to 11 am. Convective mixing was assumed to start at 9 am for the shallow boundary layer case study, 7 am for the deep boundary layer case study, and around 8 am for the case study entraining a residual layer with different properties (denoted residual in the plot) .





**Table A1.** Summary of the technical specifications of the Halo Streamline Doppler lidar.

| | |
|---|---|
| Wavelength | 1565 nm |
| Detector | heterodyne |
| Pulse repetition frequency | 15 kHz |
| Nyquist velocity | $20 \ \mathrm{m \ s^{-1}}$ |
| Sampling frequency | 50 MHz |
| Velocity resolution | $0.038 \ \mathrm{m \ s^{-1}}$ |
| Points per range bin | 10 |
| Range resolution | 30 m |
| Pulse duration | $0.2 \ \mu \mathrm{s}$ |
| Lens diameter | 8 cm |
| Lens divergence | $33 \ \mu \mathrm{rad}$ |
| Minimum range | 90 m |
| Maximum range | 9600 m |
| Telescope | monostatic optic-fibre coupled |
| Focus | 2000 m |

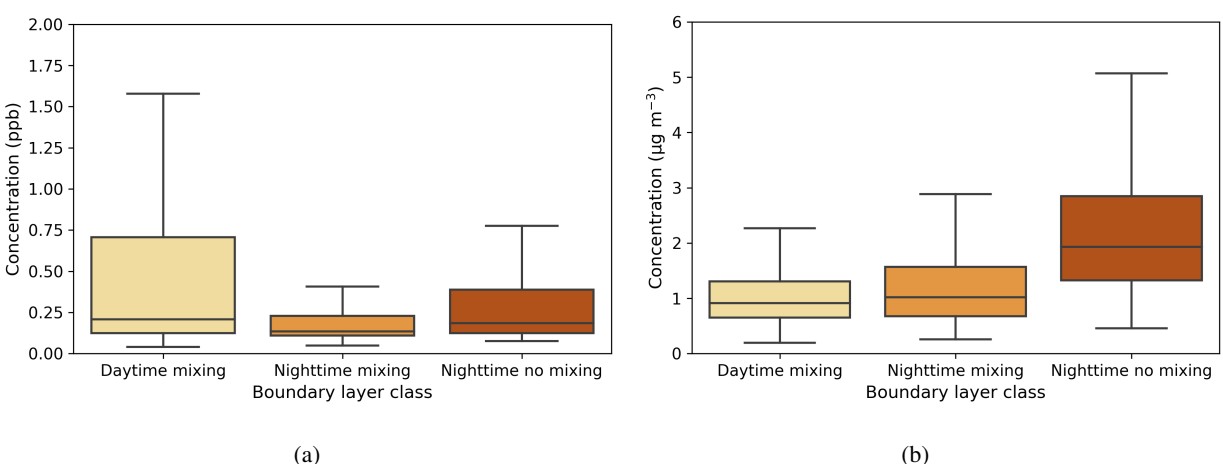

(a)

(b)

**Figure A1.** Box-and-whisker plots of hourly median values for 3 boundary layer conditions for a) $SO_2$ (ppb) and b) black carbon ($\mu g\ m^{-3}$). The central line in the boxes indicate the median, and the bottom and top lines of the boxes indicate the $25^{th}$ and $75^{th}$ percentiles. The whiskers mark the most extreme data points that are not considered as outliers.

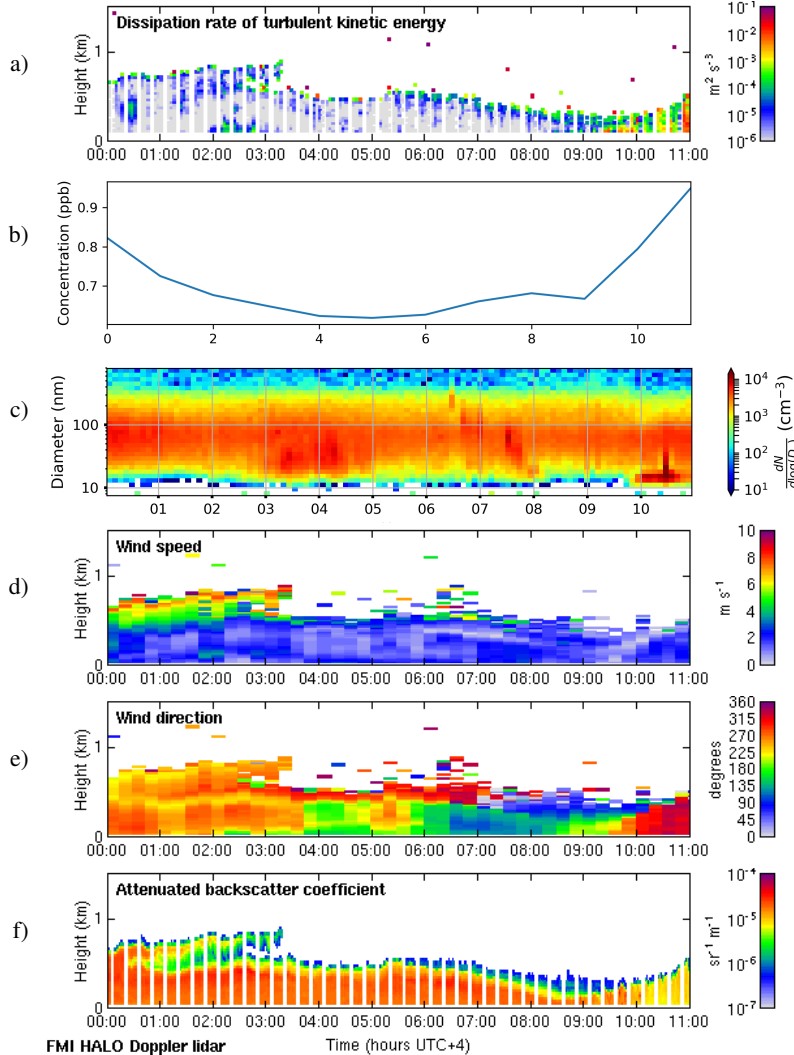

**Figure B1.** Shallow boundary layer case on 21 February 2018. Panel a) displays a time-height plot of dissipation rate of turbulent kinetic energy, $\epsilon$, b) $SO_2$ concentration (ppb) measured at the surface, c) aerosol particle size distribution measured at the surface (the color indicates the concentration $dN/d\log(d_p)$ in $cm^{-3}$), and time-height plots of d) wind speed, e) wind direction, and f) attenuated backscatter coefficient. For all panels, the time axis is local time (hours UTC+4). The measurements in panels a), d), e) and f) were obtained from the HALO Doppler lidar.

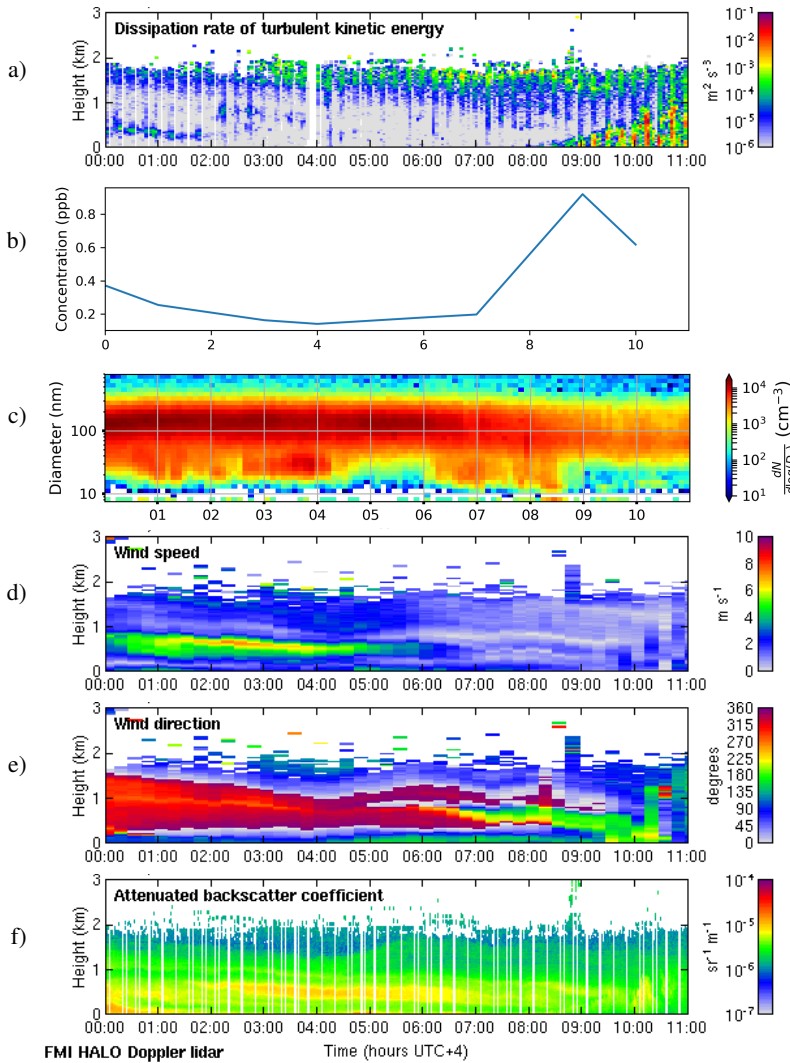

**Figure C1.** Residual layer case on 8 May 2018. Panel a) displays a time-height plot of dissipation rate of turbulent kinetic energy, $\epsilon$, b) $SO_2$ concentration (ppb) measured at the surface, c) aerosol particle size distribution measured at the surface (the color indicates the concentration $dN/dlog(d_p)$ in $cm^{-3}$), and time-height plots of d) wind speed, e) wind direction, and f) attenuated backscatter coefficient. For all panels, the time axis is local time (hours UTC+4). The measurements in panels a), d), e) and f) were obtained from the HALO Doppler lidar.