# Peer review of "Aerosol particle characteristics measured in the United Arab Emirates and their response to mixing in the boundary layer"

_Atmospheric Chemistry and Physics, 2021_

## Referee Comment (RC1)

**General**

The paper is fluently written and contains a lot of interesting results. The data quality is excellent. However, the manuscript makes an impression of a measurement report or an extended summary without a clear goal, research topic, or message.

There are so many results presented in this manuscript, but the big picture, the context, the links between meteorological conditions, dust background, sea breeze impact and anthropogenic pollution is not really obvious from all this. See the Details part for more comments.

It would be desirable to have a map of the region, and a trajectory analysis showing the main air mass transport clusters, and, finally, retrievals of the Polly lidar so that we obtain an idea about the impact of dust and non-dust (mainly pollution) aerosol components on all the in situ measurements. In this way, we would get a more complete view on the environmental and atmospheric conditions in that region of the world and even a rather modern (state-of-the-art) paper based on combined in situ surface observation and profiling observations with Polly and Halo Doppler lidars.

I have the feeling that is not so much work, therefore minor revisions are required

**Details**

P3, l81: It is stated: The Polly lidar was placed next to the container! That brings me to the question, why did you not use the backscatter and depolarization ratio profiles measured with this polarization Raman lidar? These profiles allow the separation of dust and non-dust backscatter profiles and the estimation of dust and pollution-related CCN contributions (as shown by Haarig et al., 2019, see Sect. 5.2, Figs. 7 and 8). These lidar profile data would be complementary to the high-quality ground-based in situ observations but one would be better able to quantify the contributions of dust and pollution to the detailed in situ observations. Such an approach would be a nice step forward in the field of environmental monitoring.

Haarig, M., et al., ACP, 19, https://doi.org/10.5194/acp-19-13773-2019, 2019.

P3, l83-90: This interesting paragraph on the meteorological conditions should be presented as the first subsection of Sect.3 (Observations).

P8, l233: It would be desirable to have a map with the experimental field site, oil refinieries, etc. big cities, countries. In this context, it would be desirable as well to have some typical backward trajectories, or even better, some kind of main trajectory clusters for the UAE region, for arrival heights of 500m, 1000m, 1500m. It would be interesting to see the typical wind field pattern over the day (sea breeze effects). The Halo Doppler lidar monitors such dynamical features day by day.

In the sections 3.1 and 3.1.1, many numbers are given in the text, but not as figures. It is thus not easy to handle all this information and to identify the key numbers. More visualization of results would be nice.

Sect. 3.1.2 Daily and annual variations: The question came up: Are all the findings dominated by anthropogenic pollution? What is the role or contribution of the background aerosol (dust, marine)? Again, many numbers are presented without having figures.

Sect.3.2 Halo Doppler lidar observations come into play. But then, an overview about dust and pollution layering from the Polly observations would be desirable as well. Such an overview is clearly missing. What shall we learn from Doppler lidar observations when we still have no idea about the pollution-dust mixing state as a function of height (PBL, free troposphere). By integrating Polly retrievals on dust and non-dust CCN concentrations (or other parameters) one would get a much better, more complete overview of the aerosol conditions in the UAE greater area.

P11, l339-340: As an example, you write: The reason why the activation fraction is higher…. is probably due to entrainment of the residual layer above back into the nocturnal boundary layer… But what is the aerosol in the residual layer and in the boundary layer? Only dust, a mixture of dust and pollution, or only pollution? The Polly lidar can support and help to clarify.

At the end of Sect. 3.2.1 I asked myself, what do we learn from all this. All the observations are just presented in form of a measurement report. Many data, a lot of reporting, the specific goal remains unclear.

Sect. 3.2.3 What about differences in NPF, winter vs summer?

Sect 3.3 Case studies, again a map would be helpful, trajectory clusters would be nice, Polly dust and non dust fractions …

Just to mention my main impression again: the big picture is missing based on the excellent in situ aerosol observations, Polly and Doppler lidar profile observations, trajectories, and if available, even AERONET optical and retrieved microphysical properties.

Table 2: What is the dust CCN contribution? What is the pollution CCN contribution?

Figure 4: Again, what is the dust fraction? What is the pollution fraction?

Figures 6 and 7: no seasonal differences?

Figure 8: What do we learn? Besides the impact of PBL height.

Figure 10: Without a map, typical windfields, trajectory cluster information, such a figure appears to be useless!

Figure 10f, what does the Polly lidar show?

---

## Author Response (AR1)

**Response to Reviewer comments**

**Jutta Kesti**

**September 2021**

We thank both reviewers for their positive and constructive comments. We have addressed all of the points raised by the reviewers (copied here and shown in black text), and include our responses to each point below (in blue text). Where there has been a major change in the manuscript we provide the original text (in black italics) and the new text (in blue italics).

**1   Anonymous Referee 1**

Kesti et al. present 1 year of aerosol and boundary layer data collected in the UAE. The manuscript analyzed size distributions, CCN properties and new particle formation events in the context of the evolution of the boundary layer.

Overall the manuscript is well written and easy to follow. The dataset represents a valuable contribution toward understanding aerosol properties in an undersampled region of the world. The analysis of the new particle formation events seems to be done somewhat hastily and requires more attention to detail. The conclusion that downward mixing initiates NPF at the site may be correct, but it doesn't seem to be directly shown in this work. This would require a detailed understanding on what caused nucleation in each event, which neither the authors nor anyone else really knows in the field setting. There is also the possibility that it is initiated aloft and newly formed particles are mixed down. These uncertainty should be acknowledged in the text. My main critique of the work is that it is purely observational with few tangible new scientific findings. I recommend that the manuscript is published as a "Measurement Report" instead of a "Research Article" if following comments can be addressed.

The reviewer raises very good points! As the reviewer has pointed out, we can only provide rough estimates of NPF event start times as we were not able to measure the size ranges of the smallest particles and clusters in which NPF starts. We agree that the uncertainty in growth rate estimates should be stated and have included these. We also agree that we do not know if the NPF event was initiated at the surface or aloft, only that it has started and we have observed it at the surface. We have modified the text to make this clear.

We present fundamental knowledge on the aerosol and CCN concentrations and composition in a region which has important climate implications but has been undersampled. Our holistic approach investigates and explains the aerosol

properties in the region not only in relation to their sources, but also taking into account the important, but often neglected, impact of boundary layer dynamics and transport aloft, which we can now measure at the same time. While it is true that we have described the measurement environment and presented the measured parameters and their diurnal variation quite extensively, this merely stems from the fact that previous studies, and thereby existing literature, from this region are sparse. However, the main focus of the article is to impart new scientific knowledge, such as aerosol cloud condensation properties and the role of vertical mixing on the aerosol measurements made at the surface.

**1.1 Comments**

- Please state the make, model in the text and briefly discuss the operating principle of the CCN instrument as well as the supersaturation calibration procedure.

  We have included some additional text in the manuscript:

  *A Cloud Condensation Nuclei counter (CCNc, Droplet Measurement Technology, Model CCN – 100, Roberts and Nenes, 2005) consists of a saturator unit and an optical particle counter (OPC). First, aerosol particles are brought to supersaturated conditions and after that the number of activated droplets is counted with the OPC. The CCNc was operated at a flow rate of $0.5$ l min$^{-1}$ and in five different supersaturations (0.1, 0.2, 0.3, 0.6, 1.0). Each supersaturation measurement cycle was set to take 10 minutes so one complete cycle took 1 hour. The full cycle includes an additional supersaturation of 0.09 to allow the temperatures to drop and stabilise after measuring at a supersaturation of 1.0. The CCNc system also incorporated one extra feature that is not part of the standard CCNc system provided by the manufacturer; the additional feature enabled the CCNc to measure the fraction of activated aerosol particles as a function of size by size selecting aerosol particles with a DMA (10–250 nm size range). To determine the fraction of activated aerosol particles at a certain size, the CCNc number concentration was compared to the number concentration of a CPC (TSI 3772 until end of October 2018, TSI 3310 during rest of the campaign) that was measuring in parallel with the CCNc. These scans were performed one supersaturation at a time from 10 nm to 250 nm (size ramp took 10 min) and then moving on to the next supersaturation to begin the next scan until all supersaturations were completed. The CCNc alternated between total CCN concentration and size selected CCN concentration so that every second supersaturation sequence was with a size selecting DMA in front so that each supersaturation cycle took 1 hour to complete. The CCNc was calibrated for five different temperature gradients (corresponding to $\Delta T$ of 3 – 16 °C) using an aerosol generator (TOPAS, model ATM 226) with ammonium sulphate solution (activation curve known) and a short Hauke–type DMA coupled with the CPC. The activation curve was calculated by measuring particles in a size range of 10 – 250 nm and comparing the particle counts between the CCNc and CPC. The activation curve for the different temperature gradients was used to calculate the supersaturations. After the supersaturations were calculated, a linear fit was used between the temperature gradients and calculated supersaturations, and the constants from the fit were given for the CCNc measurement program as input parameters.*

- Figures 8 and 11 should be presented as a histogram and include a measure of the range. This could be visualized either through vertical errorbars or transparent shading for the three distributions.

  Aerosol particle size distributions have typically been presented as line

plots in the literature, so we would prefer to keep these figures as line plots to be more consistent with other studies. We agree however, that it is good to show the measure of the range, so we have added shaded areas which indicate the interquartile range of the distributions. We have included additional explanatory text in the figure captions:

*The shaded areas indicate the median $\pm$ the interquartile range of the distributions.*

- There needs to be more detail given for the derivation of growth rates. Please specify the method used and show an example in a supplement.

We have added one figure (Fig. B1) in the supplement where we show 6 NPF event cases. The growth rates have been derived from visual inspection, as for our purposes, we only use these to estimate the start time of NPF events. Since we are also missing the growth rate for the smallest sizes (particles 1 – 7 nm in diameter), we use the assumption that growth rates at the smallest sizes are about 4 times slower. This results in an uncertainty of about +/- 30 minutes in our estimate of the NPF event start time. Using a more sophisticated method is unlikely to reduce this uncertainty. We have modified the text to the following:

*The actual event starting time is then estimated based on the growth rate estimated visually from 6 DMPS size-concentration figures (Fig. B1) where there was a very clear NPF event. Our estimate for the mean growth rate was approximately $6.8$ nm $h^{-1}$ $\pm$ $2.9$ nm $h^{-1}$ (one standard deviation), which is consistent with the median growth rate $7.4$ nm $h^{-1}$ observed by Hakala et al. (2019) in Saudi Arabia.*

- Please provide statistics for the growth rate.

We now provide the mean and standard deviation of the growth rate (see previous answer) and also provide the range of estimated growth rates in the manuscript:

*The visually–estimated growth rates ranged between $2.5$ and $10$ nm $h^{-1}$.*

- Applying a single growth rate to determine the start time is questionable. It needs to be derived through extrapolation from each event.

We agree that this would be preferable if we required accurate start times. For our purposes, we are comparing the start time to the time at which the boundary layer begins to grow, which also has some uncertainty as we do not quite measure to the surface. Since the growth rates at small sizes also have to be assumed, as these are not measured in this study, we think that applying a single growth rate together with appropriately large uncertainties in the derived start times is sufficient for our purposes. We have added 6 NPF event cases where the growth rate has been estimated to give a range of expected growth rates, and mean estimated growth rate $6.8$ nm $h^{-1}$ is consistent with the median growth rate $7.4$ nm $h^{-1}$ observed

by Hakala et al. (2019, https://acp.copernicus.org/articles/19/10537/2019/) in Saudi Arabia.

- Please justify the use of a single growth rate to extrapolate to the start time. Growth rates often differ in different size ranged. Provide an estimate of the uncertainty in your procedure

  This is a good point and very true. We have now estimated the growth rates for 6 different NPF cases and calculated the mean value for the growth rate. We have now stated in the text that the growth rate estimated for particles with the diameter >7 nm is not the same for smaller particles (Kulmala et al. 2004, https://acp.copernicus.org/articles/4/2553/2004/). Kulmala et al. (2004) stated that "during the typical days the observed growth rate in nucleation mode (particle diameter over 5–7 nm) are four times bigger than the growth rate for clusters (1.5–3 nm in diameter)". If we estimate a growth rate of one fourth of 6.8 nm h$^{-1}$, which is 1.7 nm h$^{-1}$, for the clusters 1.5–3 nm in diameter, we would get an approximate of 2.5 hours for particles to grow from 1 nm to 10.4 nm in size. We have modified the text to the following:

  *We subtracted 2 hours from the identification time obtained from analysing concentrations in the 10.4 nm size bin to derive the time when these aerosol particles were about 1 nm in size based on the estimated growth rate. We should highlight here that the growth rate as defined from the DMPS measurements is not directly applicable for the smallest particles, which usually have much lower growth rates (Kulmala et al., 2004). We assume that the growth rate for clusters 1.5 – 3 nm in diameter is about a quarter of the growth rate at 10.4 nm (Kulmala et al., 2004), which is 1.7 nm h$^{-1}$. Thus, we assume that it takes about 1.5 – 2.5 hours for clusters of particles 1 nm in size to grow to 10.4 nm and use this reasoning to justify the two hour delay between the probable NPF event starting time, and the event being observed with our measurement instrumentation.*

  We noticed a typo in the code which calculated the NPF event starting hours for the histogram figure 9, so the starting hours are now corrected and the text modified to the following:

  *NPF events usually started around 7 – 9 am. This was compared with the time that the boundary layer started to grow (Fig. 9b). We defined the boundary layer growth start time based on the value of the dissipation rate of turbulent kinetic energy derived from Doppler lidar, $\epsilon$, exceeding $10^{-4}$ m$^2$ s$^{-3}$ at a height of 225 meters. Boundary layer growth at this height usually started around 9 – 10 am, and if we consider that it started a little earlier at the surface, it is quite consistent with the NPF event estimated typical starting time.*

- Why was 1 nm selected as the starting point? Clusters typically activate at 3 nm or so. Please justify or change definition.

Kulmala et al. (2004, www.atmos-chem-phys.org/acp/4/2553/) stated that the initial growth at sizes 1 – 1.5 nm might be dominated by ion-mediated condensation and after that the charged clusters are neutralized and continue their growth by vapour condensation on neutral clusters. Electrically neutral critical clusters are not easy to measure due to instrumental limitations (Kulmala et al., 2006), and that is probably the reason why different studies use 3 nm size for different calculations. We have not found a study which states that clusters typically activate at 3 nm, and hence we use 1 nm as our starting point.

- The concept of downward mixing of particles/precursors and it's relationship to NPF needs to be explored in much more detail, both in the context of the literature and the observations that purportedly support the conclusion reached here. There is also a significant body of literate on the subject (See e.g. https://acp.copernicus.org/articles/18/1835/2018/acp-18-1835-2018.pdf, https://acp.copernicus.org/articles/21/7901/2021/acp-21-7901-2021.pdf and references therein)

Yes this is a very good point and would merit a separate study on its own. To address the importance of this issue, we have modified the text in the conclusion part to the following:

*The combination of instrumentation used in this campaign enabled us to identify periods when anthropogenic pollution from remote sources that had been transported in elevated layers was present, and had been mixed down to the surface. The dynamics of the vertical mixing of the aerosols and their precursors as observed here have important implications in generation of the layers that may favor or hinder new particle formation. Further studies should address the connections between vertical mixing processes and nano-size particle concentrations at similar environments.*

We also added the following text to the subsection 3.2.3 New particle formation events:

*Our explanation for the correlation between the starting time of boundary layer growth and the start of NPF events is that precursor gases from elevated levels are mixed down to the surface and initiate an NPF event at the surface, or that the NPF event is initiated aloft, which has been observed by several studies (Größ et al., 2018; Lampilahti et al., 2021; Brus et al., 2021), and the newly formed particles are then measured at the surface once they are entrained within the growing boundary layer.*

**2    Anonymous Referee 2**

The paper is fluently written and contains a lot of interesting results. The data quality is excellent. However, the manuscript makes an impression of a measurement report or an extended summary without a clear goal, research topic, or message.

There are so many results presented in this manuscript, but the big picture, the context, the links between meteorological conditions, dust background, sea breeze impact and anthropogenic pollution is not really obvious from all this. See the Details part for more comments.

We thank the referee for their comments! Also, the points raised by the referee are important. It is true that the big picture is unclear and we have modified the text to be more clear with the main goals and research topics.

It would be desirable to have a map of the region, and a trajectory analysis showing the main air mass transport clusters, and, finally, retrievals of the Polly lidar so that we obtain an idea about the impact of dust and non-dust (mainly pollution) aerosol components on all the in situ measurements. In this way, we would get a more complete view on the environmental and atmospheric conditions in that region of the world and even a rather modern (state-of-the-art) paper based on combined in situ surface observation and profiling observations with Polly and Halo Doppler lidars.

We have added a map as suggested by the referee. We have also performed trajectory analyses for the case studies as suggested by the referee, but we decided not to include the trajectory map for the typical air mass transport because this has already been done for the region by Filioglou et al. (2020, https://acp.copernicus.org/articles/20/8909/2020/). We understand that it would be nice to include information from the PollyXT lidar also, but feel that this would make the manuscript much too long. Some dust and non–dust analysis from PollyXT lidar observations at this site has already been presented in Filioglou et al. (2020) and we refer to this work. More detailed analysis from the combination of PollyXT and Doppler lidar data is planned for a future publication.

I have the feeling that is not so much work, therefore minor revisions are required.

.

**2.1 Comments**

- P3, l81: It is stated: The Polly lidar was placed next to the container! That brings me to the question, why did you not use the backscatter and depolarization ratio profiles measured with this polarization Raman lidar? These profiles allow the separation of dust and non-dust backscatter profiles and the estimation of dust and pollution-related CCN contributions (as shown by Haarig et al., 2019, see Sect. 5.2, Figs. 7 and 8). These lidar profile data would be complementary to the high-quality ground-based in situ observations but one would be better able to quantify the contributions of dust and pollution to the detailed in situ observations. Such an approach would be a nice step forward in the field of environmental monitoring. Haarig, M., et al., ACP, 19, https://doi.org/10.5194/acp-19-13773-2019, 2019.

  An article on PollyXT lidar measurements at this site has been published by Filioglou et al. (2020, https://acp.copernicus.org/articles/20/8909/2020/). Here, we refer to the results of this work when discussing the aerosol properties in elevated layers. This work also noted that that the particle depolarization ratios for the mineral dust properties over our observation site are comparable to those of African mineral dust, but that this was not the case for lidar ratios. This suggests that some more work has to be done in order to separate the dust/non–dust contributions with confidence, as we did not have a sun–photometer operating. As stated earlier, more detailed analysis from the combination of PollyXT and Doppler lidar data is planned for a future publication.

- P3, l83-90: This interesting paragraph on the meteorological conditions should be presented as the first subsection of Sect.3 (Observations).

  The paragraph has been moved to the location suggested by the referee. We created a new subsection: 3.1 Meteorological observations.

- P8, l233: It would be desirable to have a map with the experimental field site, oil refinieries, etc. big cities, countries. In this context, it would be desirable as well to have some typical backward trajectories, or even better, some kind of main trajectory clusters for the UAE region, for arrival heights of 500m, 1000m, 1500m. It would be interesting to see the typical wind field pattern over the day (sea breeze effects). The Halo Doppler lidar monitors such dynamical features day by day.

  We have added a map which includes the location of large cities, oil refineries known by the authors, and the location of the measurement site. We have also modified the text in subsection 2.1 to the following:

  *The Arabian Gulf and the city of Dubai with a population of around 3.2 million (Dubai Statistics Center) are about 70 km west from the site (Fig. 1).*

  Filioglou et al. (2020) presented back trajectory analysis for this location (their Figs. 1b and 8a). Their analyses showed that the long–range

transported aerosol particles measured at the site are mainly originating from Saudi Arabia, Iran and Iraq. We added a sentence stating this and included the reference in section 2.1:

*The long–range transported aerosol particles measured at the site are mainly originating from Saudi Arabia, Iran and Iraq (Fig. 1b, Filioglou et al., 2020).*

We are currently analysing the diurnal variability of the local wind fields, including the sea breeze effect and its impact on the atmospheric composition at the site. We hope that reviewer understands that for the paucity of space, this will be discussed more thoroughly in our future work. We have purposely selected the morning periods for our studies in order to reduce the complexity of the possible contributing and/or competing dynamical processes.

- In the sections 3.1 and 3.1.1, many numbers are given in the text, but not as figures. It is thus not easy to handle all this information and to identify the key numbers. More visualization of results would be nice.

  We have reduced the numbers given in section 3.1 and only show values when we are comparing to other campaigns, otherwise the reader is referred to Table 2. There is also one table (Table 3) and a figure (Fig. 2) about the results discussed in section 3.1.1. In the modified manuscript, we have added references to these figures and tables in the corresponding places in the text.

- Sect. 3.1.2 Daily and annual variations: The question came up: Are all the findings dominated by anthropogenic pollution? What is the role or contribution of the background aerosol (dust, marine)? Again, many numbers are presented without having figures.

  This is a very interesting question. The figures and tables which are discussed in this section are cited in the text, however, we do not have direct measurements of the sub-micron aerosol chemical composition at <24h resolution. While it is clear that fine mode aerosol composition (here analysed indirectly based on $\kappa$ parameter, N, size distribution) has a daily cycle, the aerosol anthropogenic fraction is not possible to separate with the measurement data available. In subsection 3.2.2 we also discuss the different possibilities that could explain our observations. Filioglou et al. (2020) stated that the "the measurement site is a receptor of frequent dust events, but predominantly the dust is mixed with aerosols of anthropogenic and/or marine origin". While we show results consistent with their findings, we should also note that the study by Filioglou et al. (2020) was based on PollyXT lidar data which is more sensitive to aerosol particles of accumulation mode sizes and so the aerosols studied do not directly overlap with the Aitken (<100 nm) particles which are our main focus here.

- Sect.3.2 Halo Doppler lidar observations come into play. But then, an overview about dust and pollution layering from the Polly observations would be desirable as well. Such an overview is clearly missing. What shall we learn from Doppler lidar observations when we still have no idea about the pollution-dust mixing state as a function of height (PBL, free troposphere). By integrating Polly retrievals on dust and non-dust CCN concentrations (or other parameters) one would get a much better, more complete overview of the aerosol conditions in the UAE greater area.

  We agree that integrating the PollyXT data would be ideal for understanding the relative dust/non–dust contribution, but note that there is still some work to be done to incorporate such a dataset with confidence. We have concentrated on understanding how to combine the Doppler lidar and in-situ data, specifically in the case studies for identifying the time at which the growing boundary layer begins to entrain a particular elevated residual layer. Filioglou et al. (2020) observed multiple elevated aerosol layers in the region with PollyXT and concluded that probably "up to 2 km or so, night–time residual layers contain mixtures of mineral dust and urban–marine aerosols".

- P11, l339-340: As an example, you write: The reason why the activation fraction is higher.... is probably due to entrainment of the residual layer above back into the nocturnal boundary layer... But what is the aerosol in the residual layer and in the boundary layer? Only dust, a mixture of dust and pollution, or only pollution? The Polly lidar can support and help to clarify.

  We think this is very good question. Unfortunately, the minimum range of PollyXT means that it does not measure in the stable nocturnal boundary layer, and our chemical composition measurements at the surface do not have sufficient temporal resolution (filter sampling resolution was about 96 hours). Filioglou et al. (2020) stated that the night-time residual layers up to 2 km are usually mixtures of mineral dust and urban-marine aerosols. We used $SO_2$ measurements combined with the height of turbulent mixing to determine whether the first residual layer contained anthropogenic pollution when the surface layer did not.

- At the end of Sect. 3.2.1 I asked myself, what do we learn from all this. All the observations are just presented in form of a measurement report. Many data, a lot of reporting, the specific goal remains unclear.

  We wanted to highlight the additional benefit gained from combining in-situ aerosol measurements with boundary layer turbulent mixing from the Doppler lidar. However, this is also the first time that these measurements from this campaign have been presented. We agree that we should be clear in presenting the aims of the manuscript. The following text has been added in the end of section 3.2.1:

  $SO_2$ *concentrations had their highest values during daytime mixing. More*

*pronounced differences were seen in black carbon concentrations under different boundary layer conditions; higher concentrations were observed during calm nights indicating that the source of black carbon is likely to be local and disperses in turbulent conditions. Highest median $\kappa$ values were observed during daytime mixing indicating that larger and more hygroscopic aerosol particles are present in the boundary layer during daytime. We observed the lowest CCN number concentrations during daytime when there are more newly formed nucleation mode particles present. Highest activation fractions were observed during nighttime when the boundary layer was turbulent, attributed to the entrainment of particles in the residual layer above down into the boundary layer. These results show that different boundary layer conditions affect aerosol particle properties at the site.*

- Sect. 3.2.3 What about differences in NPF, winter vs summer?

  In this study, we focused on the starting times of NPF events in order to compare this with the starting time of boundary layer growth. We note that NPF events occur all year round (four days on five on average). Differences in NPF event parameters, such as growth rates, is challenging as we did not measure the smallest particle sizes (see comments and response to reviewer one), and seasonal differences are difficult to quantify with one year of data.

- Sect 3.3 Case studies, again a map would be helpful, trajectory clusters would be nice, Polly dust and non dust fractions. . .

  A map has been added to the manuscript. Based on Filioglou et al. (2020) we can estimate the overall fraction of dust and non-dust. Unfortunately, data from Polly XT are not available for the chosen case study days, so we cannot make a detailed analysis for these days.

- Just to mention my main impression again: the big picture is missing based on the excellent in situ aerosol observations, Polly and Doppler lidar profile observations, trajectories, and if available, even AERONET optical and retrieved microphysical properties.

  We agree that we should be clear in presenting the aims of the manuscript. Unfortunately, there were no AERONET (sun–photometer) measurements available for this campaign. We have modified the text in the last paragraph of the introduction to:

  *We present the diurnal variation of aerosol particle properties, aerosol particle composition and the identification of the starting times of new particle formation events. In this study we determine how aerosol particle properties measured at the surface develop depending on the boundary layer mixing conditions.*

- Table 2: What is the dust CCN contribution? What is the pollution CCN contribution?

The dust and pollution fractions have been discussed by Filioglou et al. (2020).

- Figure 4: Again, what is the dust fraction? What is the pollution fraction?

  The comparison of in–situ aerosol particle measurements and Polly observations is not that straightforward, because Polly cannot see that well very close to the surface, and our chemical composition measurements have a resolution of 4 days. The overall fractions of dust and pollution is discussed by Filioglou et al. (2020).

- Figures 6 and 7: no seasonal differences?

  We did investigate the difference between the two seasons based on wind speeds as defined by Filioglou et al. (2020); summer between March and August, winter between September and February. There did not seem to be any major seasonal differences for the parameters we measured, and we think that a longer dataset would be needed in order to provide robust statistics. If necessary, such figures could be included in a supplement, but they would lengthen the current manuscript considerably without much additional benefit.

- Figure 8: What do we learn? Besides the impact of PBL height.

  The figure shows the impact of boundary layer height, but also explains the reason why we observe the change in CCN-activation fraction even though there is no change in the CCN concentration with increasing maximum boundary layer height.

- Figure 10: Without a map, typical windfields, trajectory cluster information, such a figure appears to be useless!

  We have added a map showing the measurement location and the pollution source regions. We have generated backtrajectories using HYSPLIT but note that these and their input meteorological datasets (0.25 degree GDAS meteorological data) do not have sufficient resolution to show all of the features observed in the Doppler lidar wind fields, particularly in the vertical dimension, and therefore, we do not know how much additional benefit they would bring to the discussion. If necessary, such figures (Fig. 1, Fig. 2, Fig. 3) could be included in a supplement.

- Figure 10f, what does the Polly lidar show?

  Unfortunately, we do not have data from PollyXT during this specific day.

**3  Noticed text typos**

- p1, line1 : an important in role in the microphysics of clouds

  Changed to *an important role in the microphysics of clouds.*

- p4, line96 (original manuscript) p8, line225 (modified manuscript) : were more pronounced

  Changed to *was more pronounced.*

- p10, line283 (original manuscript) p10, line294 : (Fig. 4d) (originally 3d)

  Changed to *(Fig. 4c).*

- p13, line396 (original manuscript) p14, line426 (modified manuscript) : a NPF event

  Changed to *an NPF event.*

[Figure]

Figure 1: Backtrajectory frequency plots for the deep boundary layer case on 19 May 2018 with backtrajectories released from 4 altitudes: a) 250 m, b) 500 m, c) 750 m, d) 1000 m. Generated using HYSPLIT using 0.25 degree GDAS as meteorological input data.

[Figure]

Figure 2: Backtrajectory frequency plots for the shallow boundary layer case on 21 February 2018 with backtrajectories released from 4 altitudes: a) 250 m, b) 500 m, c) 750 m, d) 1000 m. Generated using HYSPLIT using 0.25 degree GDAS as meteorological input data.

[Figure]

Figure 3: Backtrajectory frequency plots for the deep boundary layer case with stagnant residual layer on 8 May 2018 with backtrajectories released from 4 altitudes: a) 250 m, b) 500 m, c) 750 m, d) 1000 m. Generated using HYSPLIT using 0.25 degree GDAS as meteorological input data.